# Vascular progenitors generated from tankyrase inhibitor-regulated naïve diabetic human iPSC potentiate efficient revascularization of ischemic retina

Tea Soon Park[1], Ludovic Zimmerlin [1], Rebecca Evans-Moses[1], Justin Thomas[1], Jeffrey S. Huo[1], Riya Kanherkar[1], Alice He [1], Nensi Ruzgar [1], Rhonda Grebe[2], Imran Bhutto[2], Michael Barbato[1], Michael A. Koldobskiy[1], Gerard Lutty[2] & Elias T. Zambidis[1✉]

Here, we report that the functionality of vascular progenitors (VP) generated from normal and disease-primed conventional human induced pluripotent stem cells (hiPSC) can be significantly improved by reversion to a tankyrase inhibitor-regulated human naïve epiblast-like pluripotent state. Naïve diabetic vascular progenitors (N-DVP) differentiated from patient-specific naïve diabetic hiPSC (N-DhiPSC) possessed higher vascular functionality, maintained greater genomic stability, harbored decreased lineage-primed gene expression, and were more efficient in migrating to and re-vascularizing the deep neural layers of the ischemic retina than isogenic diabetic vascular progenitors (DVP). These findings suggest that reprogramming to a stable naïve human pluripotent stem cell state may effectively erase dysfunctional epigenetic donor cell memory or disease-associated aberrations in patient-specific hiPSC. More broadly, tankyrase inhibitor-regulated naïve hiPSC (N-hiPSC) represent a class of human stem cells with high epigenetic plasticity, improved multi-lineage functionality, and potentially high impact for regenerative medicine.

[1] Institute for Cell Engineering, Department of Oncology, The Johns Hopkins School of Medicine, Baltimore, MD, USA. [2] Wilmer Eye Institute, The Johns Hopkins School of Medicine, Baltimore, MD, USA. ✉email: ezambid1@jhmi.edu

The human retina is dependent on an intact, functional vasculature. If either the retinal or choroidal vasculature become compromised, neurons and supporting cells in ischemic areas rapidly die. During progressive diabetic retinopathy (DR), ischemic death of retinal pericytes and endothelial cells (EC)[1–4] leads to acellular vascular segments, rapid death of retinal neurons, microglial stimulation, secondary inflammation, macular edema, and subsequent retinal damage from proliferative neovascularization[5,6]. If acellular retinal capillaries could be regenerated with patient-specific cellular therapies, neuronal death and pathological neovascularization could be halted or reversed. Human induced pluripotent stem cell (hiPSC) therapies offer a versatile patient-specific approach for de novo regeneration of pericytes and EC[7,8]. Durable, albeit limited long-term in vivo engraftment of conventional hiPSC-derived vascular progenitor (VP) cells into the ischemic retina was previously reported[7]. However, despite the potential and rapid advance of regenerative medicine[9,10], vascular regenerative therapies with conventional hiPSC lines currently remain limited by highly variable differentiation efficiency and poor in vivo functionality of VP derived from them.

One critical variable impacting the differentiation efficiency and functional pluripotency of conventional hiPSC is the developmental, biochemical, and epigenetic commonality of hiPSC with primed murine post-implantation epiblast stem cells (mEpiSC), which possess a more restricted pluripotency than inner cell mass-derived mouse ESC (mESC) (Reviewed in ref. [11]). Conventional hiPSC adopt a spectrum of mEpiSC-like pluripotent states with highly variable lineage-primed gene expressions and post-implantation primed epiblast epigenetic marks that result in inconsistent or diminished interline differentiation potencies[12,13]. Moreover, epigenetic aberrations in diseased states such as diabetes further inhibit efficient donor cell reprogramming to functional pluripotent states[14–18].

Naïve hiPSC (N-hiPSC) with more primitive preimplantation epiblast phenotypes, decreased lineage priming, improved epigenomic stability, and higher functionality of differentiated progenitors may solve these obstacles, but this potential has not yet been demonstrated. Several groups have reported various complex small molecule approaches that incorporated the classical 2i cocktail (i.e., GSK3β/MEK inhibition), and putatively captured human naïve-like pluripotent molecular states that were more primitive than those exhibited by conventional, primed hiPSC (reviewed in ref. [11]). However, many of these human naïve epiblast-like states exhibited karyotypic instability, global loss of parental genomic imprinting, and impaired multi-lineage differentiation performance.

In contrast, culture of conventional hiPSC with a small molecule cocktail of LIF, the small molecule tankyrase/PARP (poly ADP ribose polymerase) inhibitor XAV939, the GSK3β inhibitor CHIR99021, and the MEK inhibitor PD0325901 (LIF-3i) avoided these caveats[12,13]. The tankyrase/PARP inhibitor-based LIF-3i method rapidly reverted conventional, lineage-primed human pluripotent stem cells (hPSC) to a stable pluripotent state that adopted biochemical, transcriptional, and epigenetic features of the human preimplantation naïve epiblast. LIF-3i reversion conferred a broad repertoire of normal, non-diseased hiPSC[12,13] with molecular characteristics that are unique to naïve pluripotency, including increased phosphorylated STAT3 signaling, decreased ERK phosphorylation, global 5-methylcytosine CpG hypomethylation, genome-wide CpG demethylation at ESC-specific gene promoters, and dominant distal OCT4 enhancer usage[12,13]. LIF-3i-reverted N-hiPSC maintained normal karyotypes, and were devoid of systematic loss of imprinted CpG patterns or irreversible demethylation defects reported in other naïve reversion systems that were attributed to prolonged culture

with MEK inhibitors[19–21]. More importantly, LIF-3i-reverted N-hPSC possessed greater differentiation potency than conventional hiPSC, and this improved functional pluripotency was directly potentiated by the inclusion of XAV939 to the classical 2i cocktail[11]. Interestingly, inclusion of XAV939 into various other stem cell growth media has now further identified a class of pluripotent stem cells with improved functionalities and expanded pluripotency capable of generating both embryonic and extra-embryonic derivatives[22–24].

In this study, we demonstrate the advantage of tankyrase/PARP inhibitor-regulated N-hiPSC for significantly improving vascular cell therapies. Conventional hiPSC reprogrammed from type-1 diabetic donor fibroblasts were stably reverted to a naïve epiblast-like state with high functional pluripotency following LIF-3i reversion. Naïve diabetic VP (N-DVP) differentiated from naïve DhiPSC (N-DhiPSC) expanded more efficiently, possessed more stable genomic integrity, and displayed higher in vitro vascular functionality than primed diabetic VP (DVP) generated from isogenic conventional diabetic hiPSC (DhiPSC). Moreover, N-DVP survived, migrated, and engrafted in vivo into the deep vasculature of the neural retinal layers with significantly higher efficiencies than isogenic primed DVP in a murine model of ischemic retinopathy. Epigenetic analyses of CpG DNA methylation and histone configurations at developmental promoters of N-hiPSC revealed tight regulation of lineage-specific gene expression and a de-repressed naïve epiblast-like epigenetic state that was highly poised for multi-lineage transcriptional activation. We propose that reprogramming of patient donor cells to tankyrase/PARP inhibitor-regulated N-hiPSC may more effectively erase epigenetic aberrations sustained from chronic diseases such as diabetes, and improve their utility in subsequent regenerative therapies.

## Results

**Naïve reversion of primed hiPSC significantly improved multi-lineage differentiation potency.** LIF-3i reversion of a broad repertoire of non-diseased conventional, primed hiPSC and human embryonic stem cells (hESC) decreased lineage-primed gene expression, and diminished the interline variability of directed differentiation typically observed amongst independent primed, conventional hPSC lines[12,13]. In contrast to reports of difficult directed differentiation of human naïve states[25,26], LIF-3i-reverted N-hiPSC differentiated directly to all three germ layer lineages with significantly higher efficiencies and more rapid kinetics of differentiation than their isogenic primed hiPSC counterparts. This high multi-lineage differentiation potency of tankyrase/PARP inhibited N-hPSC did not require either basic FGF supplementation in undifferentiated cultures, or an additional re-priming or capacitation culture step to enable differentiation competence[25,26]. For further confirmation, a cohort of isogenic (genotypically-identical) naïve vs. primed, conventional normal (non-diabetic) cord blood (CB)- and fibroblast-derived hiPSC and hESC lines with normal karyotypes (Supplementary Data 1, 2) were differentiated directly and in parallel using established multi-lineage differentiation protocols (Supplementary Fig. 1). For example, in identical neural induction differentiation conditions, N-hiPSC produced significantly higher levels of Nestin+SOX1+ neural progenitors than their isogenic primed hiPSC counterparts (Supplementary Fig. 1a). Additionally, neural ectodermal progenitor cells from N-hiPSC more efficiently differentiated into elongated neurites expressing Tuj1 than their isogenic primed hiPSC counterparts (Supplementary Fig. 1b). Similarly, LIF-3i N-hiPSC vs. their isogenic primed hiPSC counterparts were directly differentiated in parallel cultures into definitive endoderm; N-hiPSC generated significantly

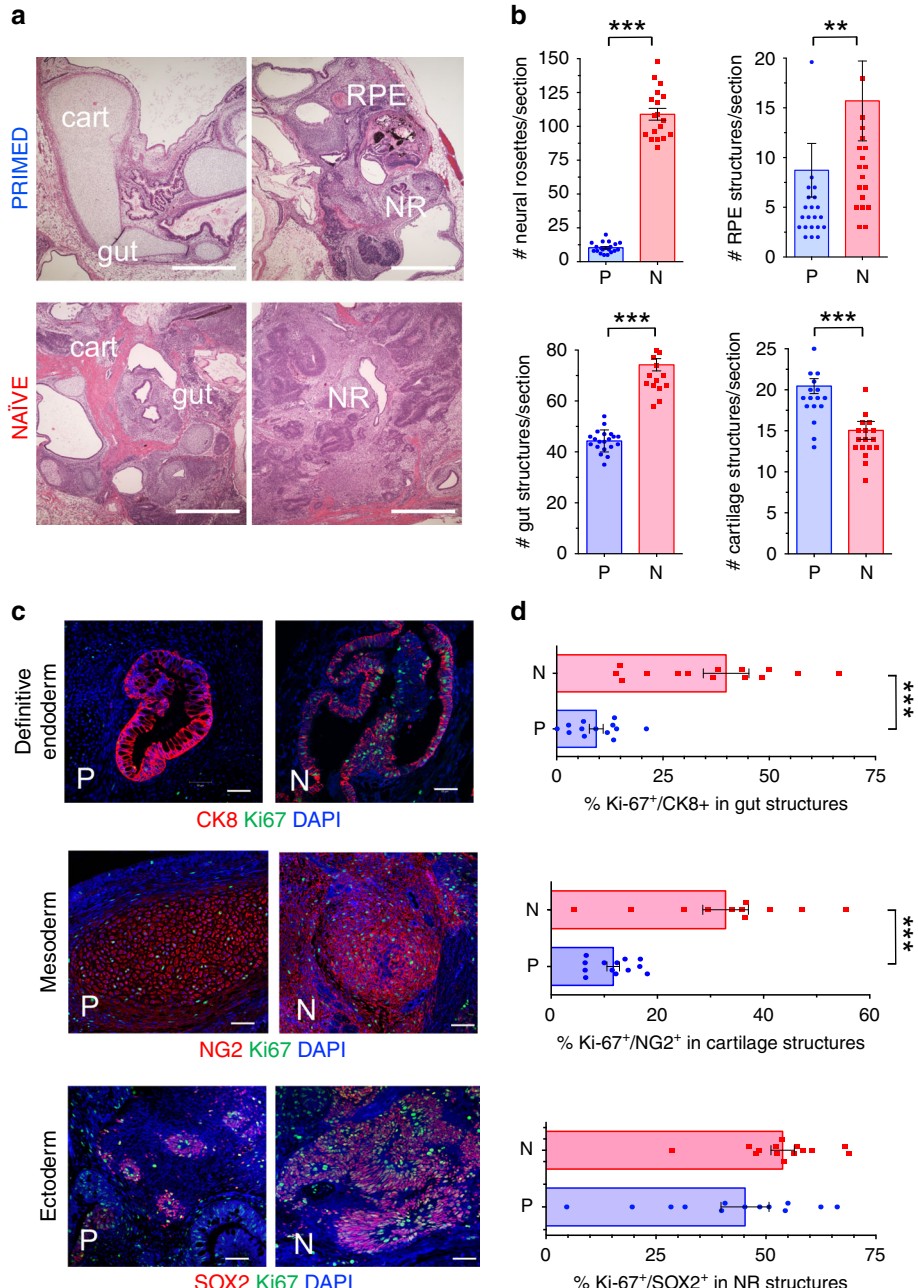

**Fig. 1 Teratoma organoid quantifications in isogenic non-diabetic hiPSC.** The non-diabetic human fibroblast-hiPSC line C1.2 (Supplementary Data 1) was cultured in parallel in either primed, conventional E8 (PRIMED; P) or LIF-3i/MEF (NAÏVE; N) conditions prior to parallel injections into sibling NOG mice ($5 \times 10^6$ cells/site) for teratoma assays. Paraffin sections of 8 week-old N vs. P teratomas were evaluated and individual microscopic sections quantified by (**a**, **b**) H&E staining (cartilage (cart); neural rosettes (NR); retinal pigmented epithelium (RPE) (Scale Bar = 500 μm), or (**c**, **d**) Immunofluorescence (IF) staining (Scale Bar = 50 μm). Shown are individual tissue section measurements from at least 3 independent teratoma experiments quantified for organoid structures and markers of endodermal (Cytokeratin 8+ (CK8); gut/glandular structures), mesodermal (NG2+ chondroblasts), and ectodermal (SOX2+ neural rosettes) lineages along with the proliferation marker Ki-67. **$p < 0.01$; ***$p < 0.001$ (Mann-Whitney tests).

higher quantities of FOXA2+ endodermal progenitors at levels surpassing isogenic primed hiPSC controls (Supplementary Fig. 1c).

To further validate the functional pluripotency of normal, non-diabetic N-hiPSC in vivo, we quantitated multi-lineage differentiation performance in teratoma assays (Fig. 1). Gross histological analysis of eight-week-old teratomas from primed and naïve normal (non-diabetic) fibroblast-hiPSC demonstrated that although both generated lineages of all three germ layers, there were marked quantitative differences between

isogenic primed and naïve-reverted fibroblast-hiPSC in generating teratoma organoid structures. Whereas primed fibroblast-hiPSC produced well-formed cystic teratomas with strong bias toward mesodermal cartilage differentiation, N-hiPSC-derived teratomas ($n = 3$) generated more homogenous and robust distribution of multiple structures from all three germ layer lineages throughout the tissues, and with significantly greater number per cross section of endodermal (gut) structures, neuro-ectodermal (neural rosettes, and retinal pigmented epithelial) structures (Fig. 1a, b). Furthermore, fibroblast-derived N-hiPSC

teratomas ($n = 3$) generated multi-lineage organoid structures with significantly higher proliferative indices (e.g., 30–50% Ki-67+ in CK8+ gut endodermal, NG2+ mesodermal cartilage structures) than their isogenic primed fibroblast-hiPSC counterpart (Fig. 1c, d).

**Tankyrase/PARP inhibitor-mediated naïve reversion of primed diabetic hiPSC.** To test the therapeutic potential of embryonic VP derived from disease-affected fibroblast-hiPSC, we generated several conventional SSEA4+TRA-1-81+ DhiPSC lines from type-1 diabetic donor skin fibroblasts using a modified version of an episomal reprogramming system[27–29] (Supplementary Fig. 2a–c). Reversion of primed normal non-diabetic hiPSC or DhiPSC with the LIF-3i naïve culture system that included the tankyrase/PARP inhibitor XAV939 (Supplementary Fig. 2d), resulted in changes from flattened (Supplementary Fig. 2b) to dome-shaped SSEA4+TRA-1-81+ N-DhiPSC colony morphologies (Supplementary Fig. 2e). The primed to naïve transition was accompanied by increased protein expressions of a panoply of naïve epiblast-specific pluripotency factors and proteins (e.g., NANOG, NR5A2, DPPA3 (STELLA), E-CADHERIN, phosphorylated STAT3 (P-STAT3), and TFAP2C[30]) (Fig. 2a–c, Supplementary Data 3). All N-DhiPSC lines possessed normal karyotypes (Supplementary Data 2, Supplementary Fig. 2e), and generated robust tri-lineage teratomas with significantly higher differentiation performances of organoid structures representing all three germ layers than their primed isogenic counterparts (Fig. 2d, Supplementary Fig. 2f), and with comparable efficiencies to non-diabetic N-hiPSC (Fig. 1).

Tankyrase 1 (TNKS1; PARP-5a) and tankyrase 2 (TNKS2; PARP-5b) are PARP-domain-containing proteins that regulate the activities of a wide repertoire of target proteins via post-translational addition of poly-ADP-ribose polymers (i.e., PARylation)[31]. To validate XAV939 modulation of tankyrase-PARP activity in N-DhiPSC, we verified its known inhibition of PARylation-mediated proteolytic degradation of key proteins targeted by tankyrase PARylation, including AXIN1 (which synergizes with the GSK3β inhibitor to stabilize the activated β-catenin complex[23]), and the tankyrase 1 and tankyrase 2 proteins themselves (which self-regulate their own proteolysis by auto-PARylation)[31]. Accordingly, chemical XAV939-mediated inhibition of tankyase 1/2 PARylation activity resulted in reduced proteolytic degradation and high accumulated levels of tankyrases 1/2 and AXIN1 in LIF-3i-reverted N-DhiPSC at levels that were comparable to non-diabetic fibroblast- and non-diabetic cord blood (CB)-derived hiPSC lines (Fig. 2e).

**Naïve reversion improved efficiency of vascular differentiation of conventional DhiPSC.** We previously identified a CD31+CD146+CXCR4+ embryonic VP population differentiated from conventional (non-diabetic) hiPSC that possessed both endothelial and pericytic functionalities[7]. These conventional hiPSC-derived VP possessed prolific endothelial-pericytic potential, and engrafted and rescued degenerated retinal vasculature following ocular ischemia-reperfusion (I/R) injury.

We employed an optimized isogenic primed vs. naïve hPSC vascular differentiation system[32,33] (Supplementary Fig. 3) and verified that LIF-3i-reverted non-diabetic N-hiPSC and hESC differentiated with significantly higher efficiencies than their isogenic primed hESC and hiPSC controls to the vascular mesoderm lineage (e.g., CD31+, CD144+, CD31+CD146+, KDR+, CD34+, CD105+, CD140b+, CD90+NG2+, and CD143+ progenitors); regardless of hiPSC genotypic background or donor source (i.e., fibroblast or cord blood derived; $n = 5$) (Fig. 3a). The high differentiation performance of N-hiPSC significantly improved the poor efficiency and interline variability of vascular differentiation routinely observed between

independent conventional fibroblast-derived hiPSC lines;[12,34,35] including CD31+CD146+ VP. Kinetic analyses of parallel vascular differentiation of N-DhiPSC vs. primed isogenic DhiPSC (Supplementary Fig. 3a) revealed similarly high expressions of vascular populations (e.g., CD34, CD31, CD146, CD144), and a more rapid decrease of pluripotency markers (e.g., SSEA4, TRA-1-81) from N-DhiPSC. CD31+CD146+ naïve diabetic VP cells (N-DVP) were also more efficiently generated from N-DhiPSC, and with similar but more rapid differentiation kinetics than for primed diabetic VP (DVP) from isogenic DhiPSC counterparts (Fig. 3b–d, Supplementary Fig. 3a). Importantly, naïve reversion permitted comparable efficiency of generation of CD31+CD146+CXCR4+ VP populations regardless of conventional hiPSC donor source (i.e., diabetic or non-diabetic). For example, naïve-reverted fibroblast-derived N-DhiPSC and non-diabetic cord blood (CB)-derived N-CB-hiPSC lines both generated similar efficiencies of VP, despite a previously reported higher VP differentiation efficiency of conventional CB-hiPSC compared to conventional fibroblast-derived hiPSC[7] (Fig. 3e).

Furthermore, following MACS enrichment of CD31+CD146+ VP from vascular differentiation cultures of non-diabetic and diabetic hiPSC, naïve and primed VP populations both displayed similar expressions of a broad array of vascular markers (e.g., CD31, CD34, CD144, and CD146) (Supplementary Fig. 4c, d). Moreover, endothelial progenitor-specific (CD31+CD105hiCD144+), pericytic (CD31+CD90+CD146+) populations[36] (Supplementary Fig. 4c), and subcellular endothelial-specific organelles (e.g., Weibel-Palade (WP) bodies, and coated pits were similarly expressed in both DVP and N-DVP; albeit with some minor differences (e.g., more abundant transcytotic endothelial channels (TEC) (Fig. 3f).

**N-DVP possessed improved vascular functionality, lower culture senescence, and reduced sensitivity to DNA damage.** Endothelium dysfunction in diabetics is characterized by poor EC survival, function, and DNA damage response (DDR). Although regenerative replacement of diseased vasculature requires high functioning cell therapies, previous studies of vascular differentiation with conventional fibroblast-hiPSC revealed poor and variable growth and expansion of vascular-lineage cells, with high rates of apoptosis and early senescence [34,35]. To evaluate endothelial functionality of naïve CD31+CD146+ VP, purified primed DVP vs. N-DVP populations were re-cultured and expanded in endothelial growth medium (EGM2). N-DhiPSC-derived N-DVP were compared to isogenic primed DhiPSC-derived DVP for in vitro endothelial functionality with Dil-acetylated LDL (Dil-Ac-LDL) uptake assays (Fig. 4a), β-galactosidase senescence assays (Fig. 4b, Supplementary 5a), quantitative Matrigel vascular tube formation (Fig. 4c, Supplementary Fig. 5b), and 5-ethynyl-2-deoxyuridine (EdU) cell cycle proliferation assays, (Supplementary Fig. 5c), These studies collectively revealed that N-DVP maintained higher endothelial Dil-Ac-LDL uptake function, more enhanced proliferation, and significantly less culture senescence than isogenic primed DVP counterparts in endothelial re-culture and expansion conditions.

To further evaluate the relative resistance of N-DVP to senescence, we probed genomic integrity maintenance by assaying for sensitivity to double stranded DNA breaks (DSBs) following treatment with the radiation damage mimetic neocarzinostatin (NCS), which triggers both DDR and pH2AX-mediated reactive oxygen species (ROS) signals[37]. Expression of phosphorylated p53 protein (P-p53), phosphorylated H2AX (pH2AX), RAD51, RAD54, phosphorylated DNA-PK (P-DNA-PK), which are all normally activated briefly following DDR and mediate repair of DSBs, were compared in re-cultured and expanded primed DVP vs. N-DVP, before and after treatment with NCS (Fig. 5a–c). These studies revealed that levels of

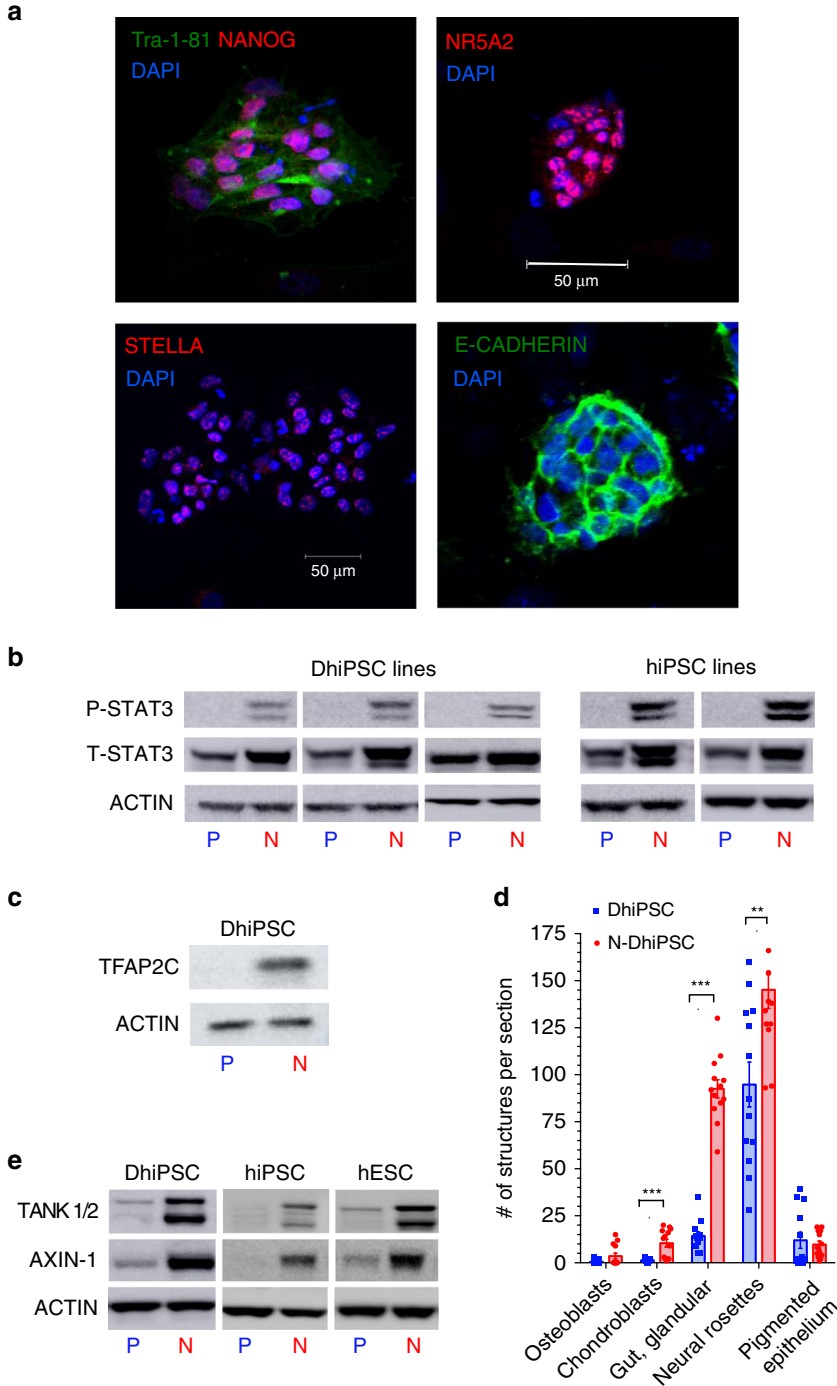

**Fig. 2 Characterization of primed vs. naïve DhiPSC. a** IF stains of N-DhiPSC (line E1C1) for general pluripotency factors (TRA-1-81, NANOG) and naïve pluripotency proteins (NR5A2, STELLA/DPPA3, E-CADHERIN; Scale Bar = 50 μm). **b** Primed (P) vs. naïve (N) phosphorylated-STAT3 (P-STAT3) expression. Western blots were performed of isogenic P vs. N lysates of (left panel) three independent DhiPSC lines (E1C1, E1CA1, E1CA2), or (right panel) two independent non-diabetic fibroblast-hiPSC lines (C1.2, C2). ACTIN and total STAT3 (T-STAT3) served as internal loading controls. **c** Naïve-specific protein expression of TFAP2C[31] in DhiPSC (E1C1) in P vs. N conditions. **d** Isogenic teratoma organoid quantifications from DhiPSC (E1C1) cultured in primed (blue bar) vs. naïve (red bar) conditions. Shown are quantifications per cross section of mesodermal (NG2+ chondroblast), definitive endodermal (CK8+ gut/glandular cells), and ectodermal (SOX2+ neural rosettes; retinal pigmented epithelium) structures from H&E stained slides. **p < 0.01; ***p < 0.001 (Mann-Whitney tests). **e** Western blots of XAV939-inhibited proteolysis of tankyrases 1 and 2 (TANK ½) and AXIN-1 proteins in isogenic primed vs. naïve conditions from DhiPSC (E1C1), and non-diabetic hiPSC (E5C3) and hESC (H9).

the DDR and DSB protein network (e.g., pH2AX, P-p53, RAD51, RAD54, and P-DNA-PK) were globally reduced in their expressions in both non-diabetic N-VP and N-DVP relative to isogenic primed VP and DVP cells (Fig. 5c, Supplementary Fig. 6); both before and

following NCS DNA damage exposure. Collectively, these data demonstrated that N-VP and N-DVP may possess an improved genomic integrity with a reduced sensitivity to stress-induced DNA damage relative to primed VP and DVP.

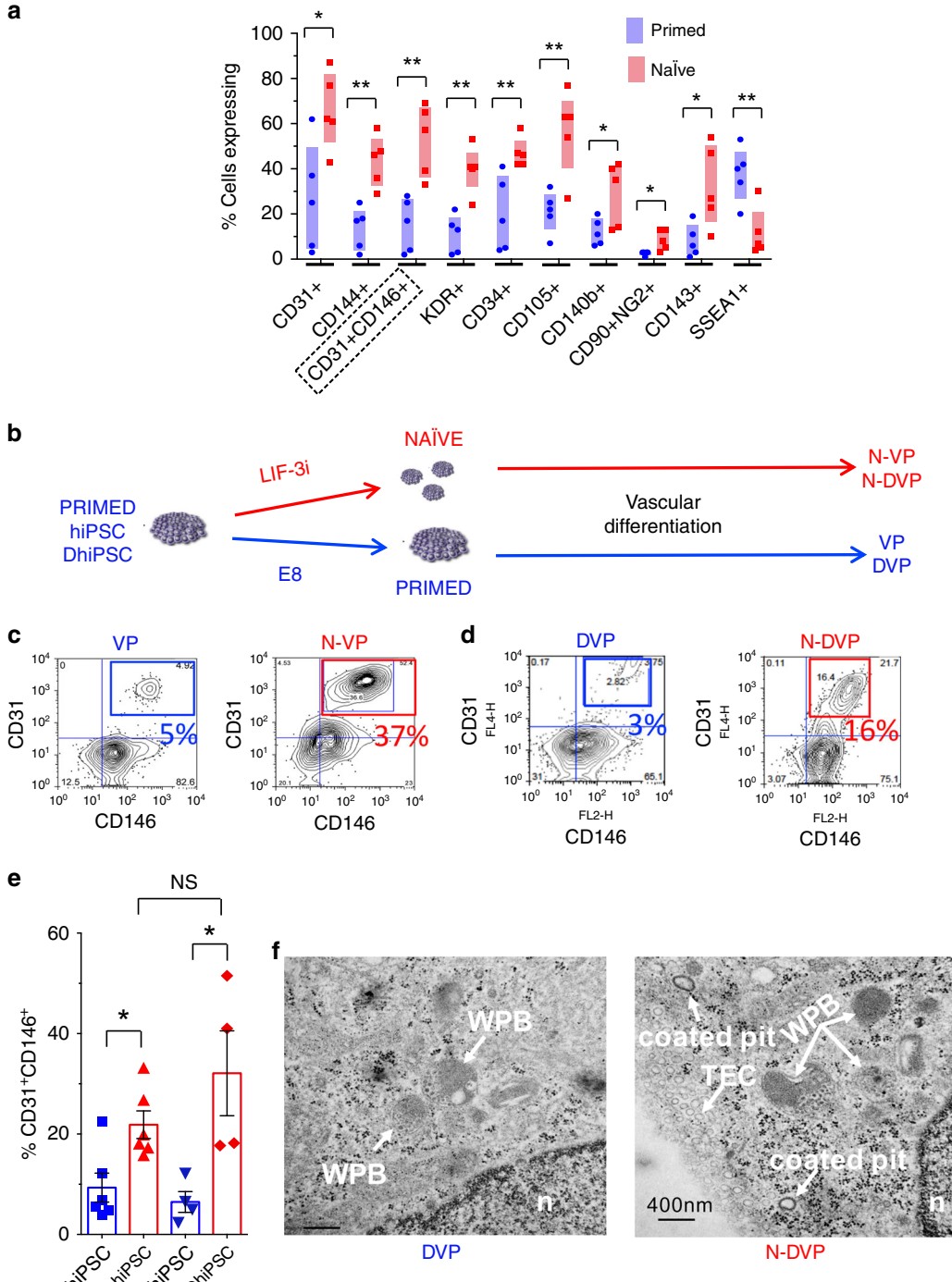

**Fig. 3 Vascular differentiations of primed vs. naïve non-diabetic and diabetic hiPSC. a** Vascular-lineage differentiation of non-diabetic isogenic primed vs. naïve hiPSC ($n = 5$ lines; Supplementary Data 1). Shown are % cells quantitated by flow cytometry on Day 10 differentiation cultures[7] expressing surface markers for endothelial-vascular progenitors (e.g., CD31+, CD34+, CD144+, CD31+CD146+, KDR+), pericytes (e.g., CD140b+, CD90+NG2+), angioblasts (e.g., CD105+, CD143+), and non-vascular-lineage markers (e.g., SSEA1+). *$p < 0.05$; **$p < 0.01$ (two-tailed unpaired $t$-tests). **b** Schematic of experimental design for CD31+CD146+ DVP differentiation of isogenic primed vs. naïve sibling hiPSC/DhiPSC. DhiPSC/hiPSC were directly differentiated (without need for an additional re-priming or capacitation step) in parallel APEL vascular conditions for 8–10 days. **c** Representative flow cytometry analyses of isogenic day 10 differentiations of (**c**) CD31+CD146+ VP vs N-VP from non-diabetic hESC (RUES02), and **d** CD31+CD146+ DVP vs N-DVP from diabetic hiPSC (E1C1). **e** Average percentages of CD31+CD146+ primed (blue) and naïve (red) VP cells obtained from isogenic day 8–10 differentiations of non-diabetic (E5C3) and diabetic hiPSC (E1CA1, E1CA2). *$p < 0.05$ (unpaired two-tailed $t$-tests). **f** TEM images of primed DVP and N-DVP differentiated and expanded from parallel primed and naïve isogenic conditions of the DhiPSC (E1C1). Weibel-Palade body (WPB), nucleus (n), transcytotic endothelial channel (TEC); Scale Bar = 400 nm.

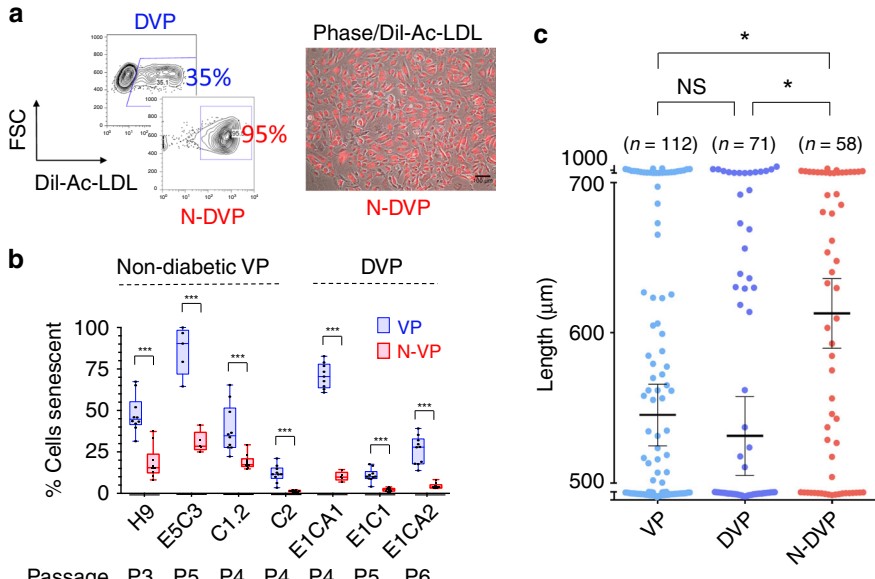

**Fig. 4 Vascular functionality of primed vs. naïve VP. a** Endothelial function. Shown are representative flow cytometry (left panel) and immunofluorescent Dil-acetylated-LDL (Dil-Ac-LDL) endothelial uptake assays (right panel); merged phase contrast/ Ac-Dil-LDL-labeled primed DVP vs. N-DVP cells; Scale Bar = 100 μm. DVP cells were generated from primed vs. naïve isogenic DhiPSC line E1CA2. **b** Expanded (non-diabetic) VP and N-VP and (diabetic) DVP and N-DVP were quantitated for senescent cells by β-galactosidase activity colorimetric assay. Shown are independent isogenic comparisons of both non-diabetic primed VP and N-VP (i.e., generated from H9 hESC, E5C3 CB-hiPSC, C1.2, C2 fibroblast-hiPSC lines) and diabetic DVP and N-DVP (i.e., generated from E1CA1, E1CA2, E1C1 fibroblast-DhiPSC lines). Each quantitation is an independent measurement of EGM2 cultures at indicated matched passages for each VP and N-VP type. ***$p < 0.001$ (multiple unpaired $t$-tests). **c** Quantification of vascular tube lengths formed from in vitro Matrigel tube assays from primed non-diabetic VP (E5C3) and isogenic primed DVP vs. N-DVP (E1CA2). The number (n) of total measurements of each of the three experimental groups from 3–5 independent experiments per group is labeled. *$p < 0.05$ (unpaired $t$-tests).

**N-DVP survived, migrated, and engrafted into ischemia-damaged neural retinal vasculature with high efficiency.** To evaluate the potential of N-DVP for in vivo engraftment and repair, we employed a humanized experimental NOG mouse model of ocular ischemia-reperfusion (I/R) injury that allows human vascular cell engraftment in an ischemic retinal niche (Fig. 6a)[7]. This rodent model has been utilized extensively to recreate the ischemic damage observed in diabetic retinopathy[2]. Intraocular pressure can be experimentally elevated, followed by allowance of vascular reperfusion. This manipulation results in loss of retinal vasculature and apoptotic death of ischemic retinal neurons ~7 days following I/R injury. Although this model does not exactly simulate the sequence of events in diabetic retinopathy, it produces similar pathology seen in ischemic retinal diseases: acellular capillaries and ischemic death of neurons. The sequential loss of murine host retinal vasculature ECs following ocular I/R injury was detected in this mouse model with an antibody specific to mouse CD31 (mCD31), and the vascular basement membrane was detected with a murine-specific anti-collagen type-IV (mCol-IV) antibody. In this model, ischemic damage is more severe in capillaries and veins than arteries, presumably due to higher collapsibility under increased intraocular pressure. Despite ischemic damage to capillaries, the basement membrane shared by EC and pericytes in retinal capillaries remains intact.

CD31+CD146+-enriched human DVP cells were differentiated from isogenic primed vs. N-DhiPSC as above, cultured briefly in EGM2, and 50,000 primed DVP or N-DVP cells were injected in parallel directly into the vitreous body of NOG recipient eyes 2 days following I/R injury (Fig. 6a). Human cell viability, intra-retinal migration, and engraftment in murine retina was evaluated at 1, 2, 3, and 4 weeks following human DVP injections with human-specific anti-human nuclear antigen (HNA) antibody staining. At 1–4 weeks following vitreous body injection in I/R-damaged eyes, HNA+ N-DVP survived in significantly greater frequencies than primed DVP within the superficial layer of the retina (Fig. 6b–d, Supplementary Fig. 7a). Additionally, HNA+ N-DVP assumed adherent abluminal pericytic and luminal endothelial positions with greater frequencies (Supplementary Fig. 7b), and appeared to favor venous engraftment of blood vessels with large diameter than arteries suggesting a preferential migration in response to injury signals[7]. In contrast, primed DVP cells at 2–4 weeks post-injection survived poorly and migrated inefficiently into ischemia-damaged blood vessels, and instead remained primarily in either the vitreous or adherent to the adjacent superficial layer of the retina (Fig. 6b–d). N-DVP not only survived significantly longer at 4 weeks post-injection robustly in the superficial retinal ganglion cell layers (GCL) in response to injury (Fig. 6c), but also engrafted into murine retinal vessels with significantly greater human CD34+ grafting efficiencies than primed DVP (Fig. 6e, f; Supplementary Fig. 7c, d).

Interestingly, further analysis of deeper retinal vessels in transverse sections of the neural retina with anti-human CD34 and anti-human CD31 antibodies confirmed significantly higher endothelial engraftment from N-DVP than from primed DVP (Fig. 7). At 2 weeks following I/R injury, human CD34+, and human CD31+ vessels were already readily detectable in N-DVP-injected retinae at significantly higher rates than primed DVP-injected retinae in the intima of murine host ischemia-damaged capillaries located in the ophthalmic artery distribution of the deep neural retinal layers. Notably, N-DVP efficiently migrated from the outer vascular layers of the GCL to form engrafted chimeric CD34+ and CD31+ human vessels in the deeper outer

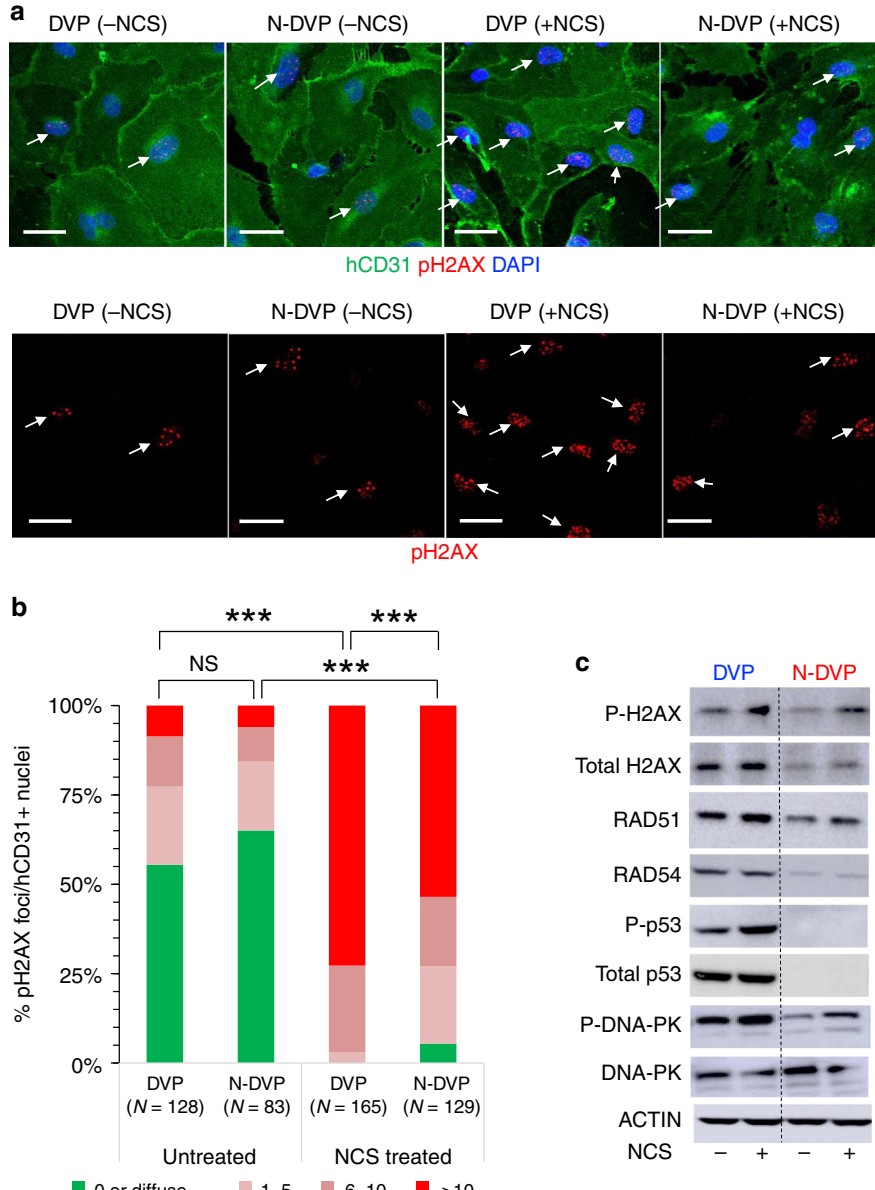

**Fig. 5 DNA damage response of primed vs. naïve DVP. a** purified and expanded DVP or N-DVP (E1CA2) were treated with the radiomimetic drug NCS for 5 h before fixation and staining with antibodies for detection of human CD31+ cells and phosphorylated H2AX (pH2AX) positive nuclear foci (i.e., DAPI co-staining) to reveal double-strand DNA breaks (arrows); Scale Bar = 50 μm. **b** Quantification of pH2AX foci per nuclei in isogenic DVP vs. N-DVP with or without induction of NCS DNA damage. Shown are numbers of DAPI+ nuclei per field with no pH2AX foci (green), and DAPI+ nuclei with 1–5 foci (light pink), 6–10 (dark pink) and >10 pH2AX foci (red). ***$p < 0.0001$; Chi-Square tests (**c**) Lysates of primed DVP and N-DVP (E1C1) cultured in EGM2 and treated with (+) or without (−) NCS were analyzed by Western blotting for expressions of proteins activated by DNA damage and apoptosis (i.e., total H2AX and phosphorylated H2AX (P-H2AX), RAD51, RAD54, phosphorylated p53 (P-p53), total DNA-PK, and phosphorylated DNA-PK (P-DNA-PK).

plexiform layers (OPL), inner nuclear layers (INL), and inner plexiform layers (IPL) of the murine neural retina (e.g., ~5–20 CD34+ human-murine chimeric vessels per 450 μm cross-section areas; Fig. 7a–d, Supplementary Fig. 8a, b). In contrast, injected primed DVP-derived human CD34+ and CD31+ cells were detected primarily in the superficial vascular layers of the retinal GCL or the inner limiting membrane (ILM); without further significant migration into deeper neural layers (Fig. 7b–d). Collectively, these data demonstrated that N-DVP but not primed DVP migrated more efficiently from vitreous into the injured deep vascular neural retinal layers; suggesting that an efficient reparative injury-induced human-murine chimeric

vasculogenesis had occurred following N-DVP (but not primed DVP) cellular therapy.

**N-DhiPSC were configured with de-repressed bivalent histone marks at key developmental promoters.** The murine naïve pluripotent state has higher differentiation potential than the primed murine pluripotent state[11,38], and is distinguished by chromatin poised for unbiased gene activation[38], global reduction of CpG DNA methylation[39], and decreased repressive H3K27me3 histone deposition at bivalent Polycomb repressor Complex 2 (PRC2)-regulated promoter sites[40,41]. To explore the molecular mechanisms

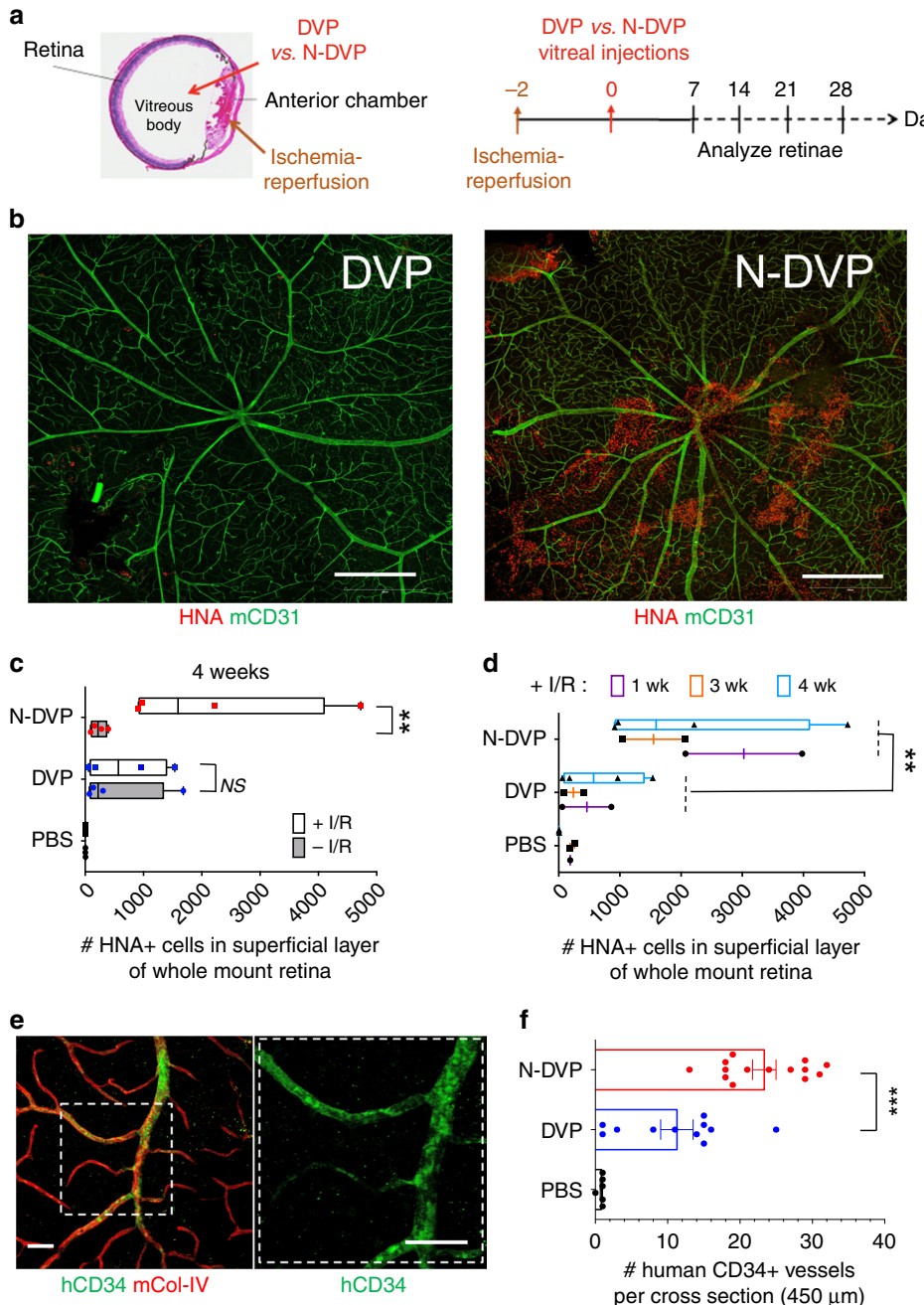

**Fig. 6 Survival and vascular engraftment of primed vs. naïve DVP in I/R-injured murine retinae. a** Schematic of NOG mouse ocular I/R experimental system for testing in vivo functionality of human primed DVP vs. N-DVP (modified from Park et al., 2014[7]). (left) Anatomical structures where I/R (anterior chamber) and human DVP and N-DVP cell injections (vitreous body) were performed (left panel). (right) Timeline for I/R injury surgery, human DVP injections (Day 0), human cell survival and engraftment analysis. **b** Human DVP survival at the superficial layer of murine retina at 3 weeks following injection of 50,000 DVP or N-DVP into the vitreous of I/R-treated NOG mouse eyes. Flat whole-mounted retinae were stained with antibodies for human-specific HNA (red), and tile scanned by confocal microscopic imaging (10x objective, 9 × 9 tiles). Shown are representative whole retinal images with HNA⁺ cells from primed DVP cell-injected (left panel) vs. N-DVP cell-injected (right panel) eyes. Scale bars = 500 μm. **c** Quantitation of HNA⁺ cells detected in the outer superficial layers of whole mount retinae following treatment of eyes with and without I/R, and injected with either primed DVP or N-DVP at (**c**) 4 weeks or (**d**) at 1, 3, and 4 weeks following DVP vs. N-DVP vs. control saline (PBS) injections, in eyes treated with and without I/R injury. Shown are the mean numbers from independent eye experiments of total HNA⁺ cells counted with imaging software per superficial layer of each whole-mounted retinae (whole field). **p < 0.01 (Mann-Whitney tests). **e**, **f** Human vascular engraftment into murine vessels. Whole-mounted retinae of I/R-injured eyes 2 weeks following DVP vs. N-DVP vs. PBS injections were immunostained with human CD34 (hCD34) to detect human endothelial engraftment. Antibodies for murine collagen type-IV (mCol-IV) were also employed to detect murine blood vessel basement membrane, and murine CD31 (mCD31) to detect murine endothelium. **f** The number of CD34⁺ human-murine chimeric vessels per 450 μm cross-section was quantitated via confocal microscopy and imaging software. Shown are results of independent measurements. ***p < 0.001 (unpaired t-tests).

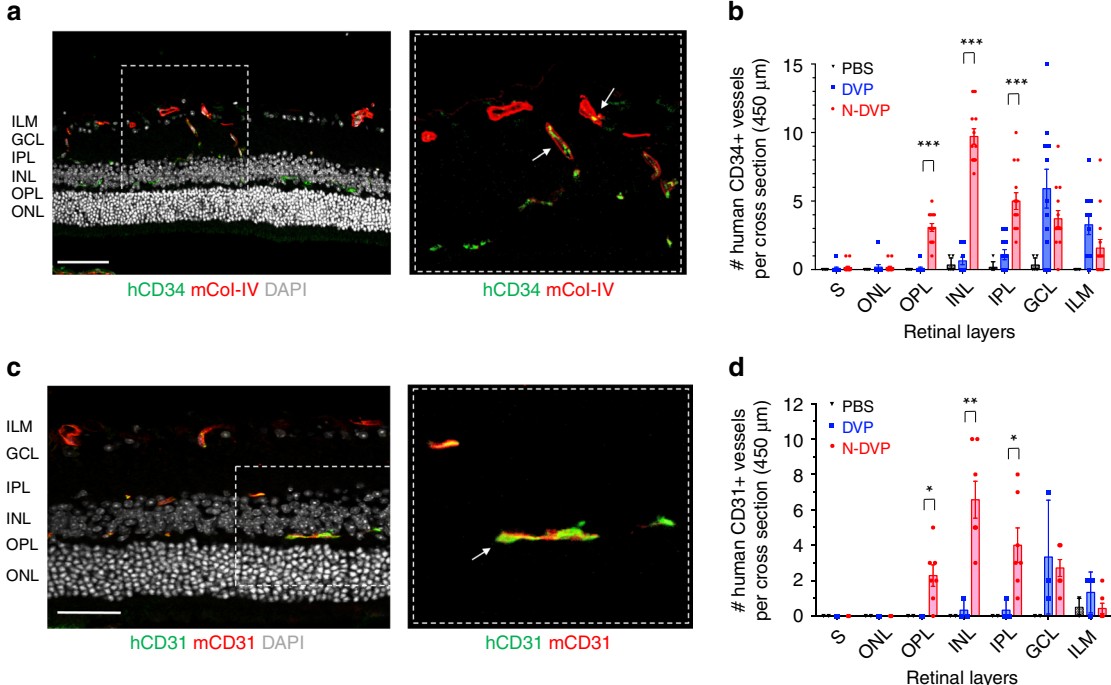

**Fig. 7 Migration and vascular engraftment of primed vs. naïve DVP into ischemia-injured blood vessels of the neural retina.** Cross-sectioned retinae of I/R-injured eyes of NOG mice were immunostained with either human CD34 (**a**, **b**; hCD34) or human CD31 (**c**, **d**; hCD31) antibodies 2 weeks following DVP vs. N-DVP injections. Antibodies for murine collagen type-IV (mCol-IV) were also employed to detect murine blood vessel basement membrane, and murine CD31 (mCD31) detected murine endothelium. The number of **b** human CD34$^+$ or **d** human CD31$^+$ cells detected within transverse layers of the murine neural retina (per 450 μm retinal cross section) was quantitated with imaging software. Each data point represents a replicate individual 450 μm retinal cross section that was analyzed from I/R- treated eyes injected with saline (PBS), primed DVP or N-DVP. Human CD34$^+$ or human CD31$^+$ endothelial cell engraftment was enumerated in each distinct layer of neural retina shown; only N-DVP migrated into the inner nuclear layer (INL) while most of the primed DVP remained primarily in the superficial ganglion cell layer (GCL). ILM: inner limiting membrane, IPL: inner plexiform layer, outer nuclear layer (ONL), OPL: outer plexiform layer, S: segments. All scale bars = 50 μm. Each individual quantitation shown is an independent experimental measurement with (standard error of mean) SEM from an individually-immunostained cryosection for each of the three groups of injected mouse eyes (i.e., saline-PBS ($n = 8$), DVP ($n = 3$, CD31; $n = 11$, CD34), N-DVP ($n = 7$, CD31; $n = 14$, CD34). ***$p < 0.001$; **$p < 0.01$; *$p < 0.05$ (multiple unpaired $t$-tests).

that drove improved vascular functionality of LIF-3i-reverted N-DhiPSC, we probed the global transcriptional profile and epigenetic configurations that may potentially regulate a more faithful vascular gene expression in N-DVP. We evaluated the global transcriptomic profiles of naïve vs. primed normal and diabetic VP, as well as their parental hPSC lines by performing RNA-sequencing (RNA-Seq; Supplementary Data 4, 5). Principal component analyses (PCAs) of VP/DVP vs. N-VP/N-DVP transcriptomes demonstrated that both normal and diabetic samples possessed sharply distinguished global expression signatures (Fig. 8a). A differential gene expression analysis revealed that naïve VP from both normal and diabetic sources were enriched in hemato-vascular stem-progenitor genes (e.g., *ANGPT1, MMP9, VEGF, ITGB1, HOXA11, KLF1*), and pathways (e.g., KLF1, RUNX1, STAT5A, GATA1 gene targets) (Fig. 8b–d, Supplementary Data 4).

Additionally, all three fibroblast-N-DhiPSC lines exhibited significant reductions in global 5-methylcytosine (5mC)-associated CpG DNA methylation following LIF-3i reversion (Fig. 9a), similar to non-diabetic fibroblast-N-hiPSC[12]. The detection of increased global levels for 5-hydroxymethylcytosine (5hmC) in N-DhiPSC relative to primed DhiPSC further suggested a potential contribution of naïve-like TET-mediated active CpG demethylation[39].

Gene-specific enrichment analysis (GSEA) of RNA-Seq of VP samples revealed that primed VP were enriched in non-vascular-specific lineage genes (e.g., neuron-specific PRC2 gene targets) relative to N-VP. This suggested that lineage priming in

conventional hPSC had affected not only the efficiency but also the epigenetic fidelity of vascular differentiation in primed VP (Fig. 8d). To better elucidate the epigenetic mechanisms regulating decreased lineage priming, simultaneous bioinformatics analyses of both expression and CpG methylation of lineage-specifying PRC2 gene targets was performed (before and after LIF-3i-reversion of a broad array of isogenic hiPSC lines). This analysis revealed a broad rewiring of the lineage-specifying machinery. CpG DNA hypomethylation at promoter sites of PRC2 genes were broadly hypomethylated in N-hiPSC lines relative to their primed counterparts with a significant decrease of expression in many lineage-specifying PRC2 targets (Supplementary Fig. 9a, b). Collectively, these studies revealed that in comparison to primed fibroblast-hiPSC, isogenic N-hiPSC displayed significantly less baseline epigenetic CpG methylation-regulated repression of lineage-specifying PRC2 genes, despite a broad silencing of lineage-primed transcriptional targets of PRC2 as previously reported[12] (Supplementary Fig. 9c, Supplementary Data 5).

A critical mechanism for preventing mouse ESC from lineage priming is via regulating the poised silencing or activation of lineage-specifying genes at PRC2-associated bivalent H3K27me3 repressive and H3K4 activation histone marks, and RNA Polymerase II (POLII) pausing at promoter sites[40–42]. Thus, we next assessed the protein abundance of PRC2 components which mediate repressive H3K27me3 deposition on bivalent promoters in naïve versus primed normal and DhiPSCs (Fig. 9b). Interestingly,

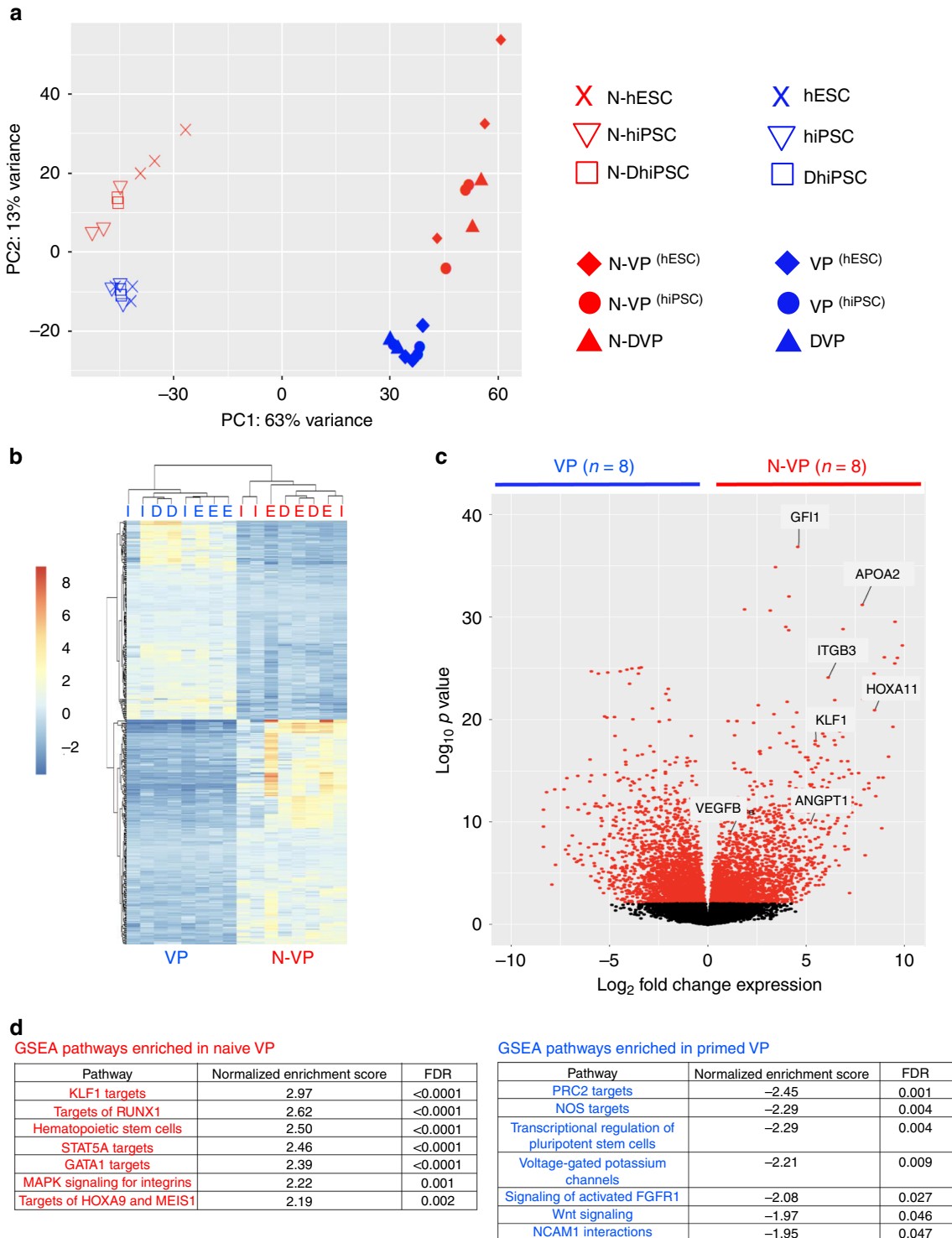

**Fig. 8 Transcriptional profiling of primed vs. naïve VP from normal and diabetic hPSC. a** PCA of whole genome transcriptomes from RNA-Seq samples from primed VP/DVP vs. N-VP/DVP, and their parental isogenic hPSC lines (i.e., primed or naïve hESC-derived VP, primed or naïve hiPSC-derived VP, and primed or naïve DhiPSC-derived DVP). **b** Heatmap-cluster dendrogram of the top 500 most differentially expressed genes (Supplementary Data 4); hierarchical clustering (Euclidian distance) of isogenic primed vs. naïve VP RNA-Seq samples. Isogenic paired VP samples are same above (n=8; VP/DVP and N-VP/N-DVP). (hESC-derived VP or N-VP: 'E'; hiPSC-derived VP or N-VP: 'I' ; DhiPSC-derived DVP or N-DVP: 'D'). **c** Volcano plot of differentially expressed transcripts in whole genome of primed vs. naïve VP; log$_{10}$ p-values vs. log$_2$ fold change in expressions. RNA-Seq VP samples are same as in PCA above (n = 8). **d** GSEA of pathways enriched in primed VP/DVP vs. N-VP/N-DVP. Paired isogenic primed and naive VP samples used for analysis are same as PCA (n = 8).

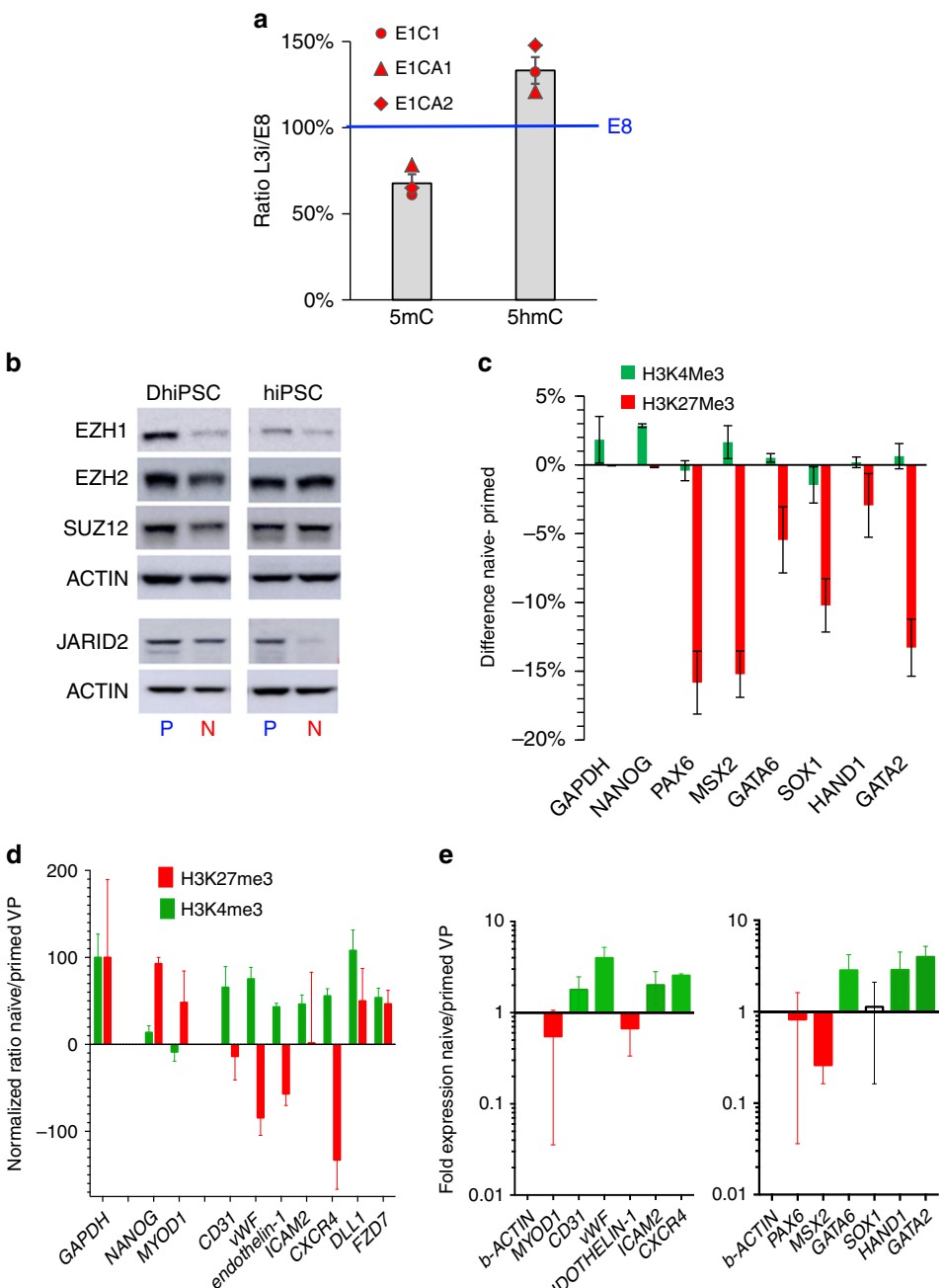

**Fig. 9 Epigenetic configuration of bivalent and vascular-lineage-specific promoters in primed vs. naïve hiPSC and VP. a** Densitometric quantitation of dot immunoblots of genomic DNA samples for global levels of 5-methylcytosine (5mC) and 5-hydroxymethylcytosine (5hmC) in three isogenic pairs of primed vs. naïve DhiPSC. Immunoblot naïve/primed densitometric ratios were determined with ImageJ software at steady state conditions (200 ng), and normalized at 100% for E8 values. **b** Western blot analysis of PRC2 components (EZH1, EZH2, SUZ12, and JARID2) in primed vs. naïve normal (C1.2) and diabetic (E1C1) hiPSC lysates. **c** ChIP-qPCR for H3K27me3 and H3K4me3 histone marks at bivalent developmental promoters (e.g., PAX6, MSX2, GATA6, SOX1, HAND1, GATA2) in primed vs. naïve DhiPSC (E1C1). GAPDH and NANOG are controls for actively-transcribed genes. Data are presented as differences in percent input materials of naïve minus primed genomic DNA samples. Error bars represent the SEM of replicates. **d** ChIP-qPCR for H3K27me3 and H3K4me3 histone marks at vascular developmental promoters in primed vs. naïve VP genomic samples. Data are presented as GAPDH-normalized ratios of percent input materials between naïve and primed VP differentiated from DhiPSC (E1C1). Results are shown as ratios of expression of isogenic N-DVP vs. DVP for GATA2-regulated genes (CD31, vWF, endothelin-1, ICAM2) and genes regulated by histone marks that are known to effect vascular functionality (CXCR4, DLL1, FZD7). GAPDH served as housekeeping control gene; NANOG and MYOD1 represented control promoters that are normally repressed during vascular differentiation. **e** qRT-PCR gene expression analysis of vascular-lineage genes (left panel) and PRC2-regulated lineage-specific genes (right panel) in DVP vs. N-DVP that were differentiated from isogenic pairs of naïve vs. primed D-hiPSC (*n* = 3; E1C1, E1CA1, E1CA2). Fold changes are normalized to beta-actin expression.

these studies revealed significantly decreased abundance of multiple components of the PRC2 complex in both diabetic and non-diabetic N-hiPSC, including the enzymatic subunits EZH1, EZH2, and the cofactor subunit JARID2[43] (which drives the localized recruitment of PRC2 at developmental promoters in mouse ESC). To functionally validate the activity of PRC2 targets, we employed chromatin immunoprecipitation followed by qPCR (ChIP-qPCR) on previously characterized lineage-specific bivalent gene promoters (e.g., *PAX6, MSX2, GATA6, SOX1, HAND1, GATA2*; (Fig. 9c, Supplementary Fig. 9d) to investigate the levels of bivalent active (H3K4me3) and repressive (H3K27me3) histone marks at these key lineage-specifying promoters. These studies revealed significant H3K27me3 reductions (5–15% from isogenic primed E1C1 and E1CA1 DhiPSC lines) following LIF-3i naïve reversion.

Collectively, these CpG DNA methylation and histone mark studies revealed a relatively de-repressed naïve epigenetic state in N-hiPSC that appeared more poised for activation than primed DhiPSC; with a potentially decreased barrier for multi-lineage gene activation relative to primed DhiPSC. Thus, as previously reported in naïve murine ESC[38,40], despite a tighter regulation of leaky lineage-primed gene expression that was presumptively silenced through alternate naïve-like epigenetic mechanisms of bivalent promoter repression (e.g., promoter site RNA POLII pausing[40]), N-hiPSC appeared poised with a lower epigenetic barrier for unbiased multi-lineage differentiation.

**N-DVP possessed vascular epigenetic de-repression and reduced non-vascular-lineage-primed gene expression**. To determine downstream impacts of a naïve epigenetic state with lower barriers for vascular-lineage activation, we investigated the epigenetic configurations of vascular-lineage-specific gene promoters in differentiated DVP and N-DVP by ChIP-qPCR. We selected the promoters of genes regulated by the PRC2-regulated factor GATA2, which promotes expression of genes of endothelial-specific identity and function (e.g., *CD31, vWF, endothelin-1,* and *ICAM2*). We also selected promoters of genes known to be activated by chemical EZH2 and histone deacetylase (HDAC) de-repression in human endothelial progenitor cells (EPC) (e.g., *CXCR4, DLL1,* and *FZD7*)[44]. CD31⁺CD146⁺ DVPs vs. N-DVP were MACS-purified, briefly expanded in EGM2, and ChIP-qPCR was performed on promoter sites of these genes. Strikingly, relative to primed DVP, N-DVP displayed significantly increased marks for epigenetic activation (H3K4me3) and simultaneously reduced marks of promoter repression (H3K27me3) (Fig. 9d) for genes determining vascular functionality (e.g., *CD31, vWF, endothelin-1, ICAM2,* and *CXCR4*). Importantly, repressive H3K27me3 marks on N-DVP were increased relative to primed DVP for the non-vascular-lineage muscle-specific promoter MYOD1. qRT-PCR expression analysis of these transcripts confirmed that naïve VP indeed expressed significantly higher levels of these vascular genes and lower levels of non-vascular genes (e.g., *PAX6, MSX2, MYOD1*) (Fig. 9e). These results were consistent with an improved epigenetic state in N-DhiPSC that potentiated a lower transcriptional barrier for generating N-DVP with higher vascular-specific gene expressions, decreased non-vascular-lineage-primed gene expressions, and presumptively greater functionality.

## Discussion

Tankyrase/PARP inhibitor-regulated N-hiPSC represent a new class of human stem cells with improved multi-lineage functionality. In contrast, conventional hiPSC cultures adopt transcriptomic, epigenetic, and signaling signatures of lineage-primed pluripotency, and display a heterogeneous propensity for lineage bias and differentiation. Our previous studies demonstrated that embryonic VP derived from conventional CB-derived hiPSC generated with higher and more complete reprogramming efficiencies had decreased lineage-primed gene expression and displayed limited but long-term regeneration of degenerated retinal vessels[7] than conventional skin fibroblast-derived hiPSC lines with higher rates of reprogramming errors and lineage-primed gene expression, and poorer vascular differentiation. Here, we demonstrated that CD31⁺CD146⁺ endothelial-pericytic N-DVP were more efficiently generated from N-DhiPSC than from conventional DhiPSC. We demonstrated that N-VP differentiated from both normal and diabetic patient-specific N-hiPSC maintained higher functionality than VP generated from conventional, primed hiPSC. Embryonic N-VP with prolific endothelial-pericytic potential and improved vascular functionality for re-vascularizing ischemia-damaged tissues can be generated in unlimited quantities and injected at multiple target sites for multiple treatments and time periods. Such epigenetically plastic N-VP are non-existent in circulating adult peripheral blood or bone marrow. For example, adult EPC are limited in multipotency, expansion, homing, and functionality in diabetes[2,14–16]. The generation of embryonic N-DVP from a diabetic patient-specific N-DhiPSC bypasses this obstacle. N-DhiPSC are more effectively reprogrammed from a donor's skin or blood cells back to a pre-diseased state, and could subsequently be differentiated to unlimited quantities of pristine, transplantable N-DVP. Unlike adult diabetic EPC, N-DVP generated from tankyrase/PARP inhibitor-regulated N-DhiPSC would be unaffected by the functional and epigenetic damage caused by chronic hyperglycemia.

Our study has demonstrated a model system for overcoming the epigenetic obstacles of incomplete reprogramming, lineage priming, and disease-associated epigenetic aberrations in conventional hiPSC via molecular reversion to a more primitive, unbiased epigenetic configuration (Fig. 10a). Compared to conventional lineage-primed DhiPSC, tankyrase-inhibited N-DhiPSC possessed a de-repressed naïve epiblast-like epigenetic configuration at bivalent developmental promoters that was highly poised for non-biased, multi-lineage lineage specification, in a manner akin to naïve murine ESC [38–41] (Fig. 10b).

Zimmerlin et al. reported that LIF-3i reversion of conventional hPSC was sufficient for enhancing developmental potency to all three embryonic germ layer lineages[12]. Interestingly, several recent studies also incorporated tankyrase/PARP inhibition into their small molecule cocktails to significantly improve the functionality of either murine PSC[22], or alternatively human PSC[24,45] with a non-naïve epiblast-like epigenetic phenotype. For example, the LCDM[45] and mouse expanded potential stem cell (EPSC)[22] methods incorporated tankyrase or other PARP inhibitors in their chemical cocktails to improve both trophectoderm and embryonic contribution of mESC into murine chimeras. Both systems utilized tankyrase/PARP inhibition either at the initiation[45] or throughout[22,24] PSC expansion. Remarkably, tankyrase/PARP inhibition of mESC and hPSC preserved the capacity of cleavage-stage murine blastomeres for trophectoderm/ICM lineage segregation. Supplementation with XAV939 to other human naïve-like pluripotent states prior to directed differentiation also led to a significant reduction of lineage-primed gene expressions, and partially rescued a multi-lineage differentiation block[46].

Unlike the original LIF-3i method, these XAV939-inhibited human PSC methods did not continuously incorporate a MEK inhibitor in their initial growth conditions, which may be necessary for synergy to epigenetically potentiate and maintain a naïve epiblast-like pluripotent state[38,40]. LIF-3i was sufficient for bulk, stable reversion and expansion of a large repertoire of conventional lineage-primed hiPSC to a human naïve pluripotent state that possessed characteristics of human preimplantation ICM and mESC, including high clonal proliferation (i.e., without

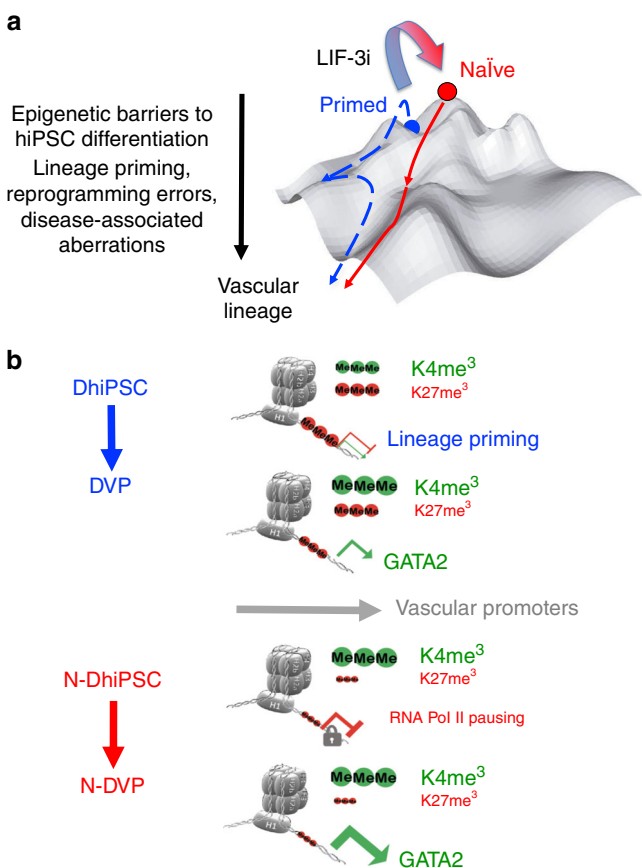

**Fig. 10 Epigenetic model for improvement of multi-lineage differentiation by reversion of primed hiPSC to a naïve pluripotent state. a** Waddington landscape model[70] for the epigenetic barriers posed by lineage priming, incomplete reprogramming, and disease-associated epigenetic aberrations in primed hiPSC (blue). These obstacles may be overcome with molecular reversion to a tankyrase/PARP inhibitor-regulated naïve epiblast-like state (red) possessing a developmentally naïve epigenetic configuration. **b** Compared to lineage-primed DhiPSC, N-DhiPSC possessed a de-repressed naïve epiblast-like epigenetic configuration at bivalent PRC2-regulated developmental promoters that was highly poised for non-biased, multi-lineage lineage specification.

requirement of apoptosis inhibition; Rock/Rho kinase inhibitor), MEK-ERK/bFGF signaling independence, activated JAK/STAT3 phosphorylation/signaling, expression of human preimplantation epiblast-like core pluripotency circuits, distal OCT4 enhancer usage, global DNA CpG hypomethylation, and increased expressions of activated β-catenin. Moreover, tankyrase/PARP inhibitor-based naïve reversion produced N-hiPSC that retained normal patterns of CpG methylated genomic imprints, reduced lineage-primed gene expression, improved multi-lineage differentiation potency, and did not require reversion culture back to primed conditions prior to differentiation.

The mechanism by which the tankyrase/PARP inhibitor XAV939 stabilized and expanded the functional pluripotency of an inherently unstable human naïve state in 2i conditions currently remains incompletely defined. However, we noted that CpG DNA methylation and histone configurations at developmental promoters of diabetic N-hiPSC possessed tight regulation of lineage-specific gene expression and a de-repressed naïve epiblast-like epigenetic state that was highly poised for multi-lineage transcriptional activation. The LIF-3i chemical cocktail employs MEK inhibition (PD0325901) to block lineage-primed differentiation along with simultaneous and

parallel XAV939 and GSK3β (CHIR99021) inhibition; which both modulate WNT[12]. Canonical WNT signaling is determined by a balance between activated non-phosphorylated β-catenin and GSK3β-potentiated phosphorylation by its destruction complex (AXIN/APC/GSK3β). Inhibition of GSK3β kinase activity impedes β-catenin destruction, and potentiates its activated non-phosphorylated form to increase the targeting of pluripotency and differentiation-associated nuclear factors. In contrast, sole use of XAV939 results in degradation of β-catenin WNT signaling via AXIN-mediated stabilization of the β-catenin destruction complex. However, when the two inhibitors XAV939 and CHIR99021 are combined simultaneously, they synergistically reconfigure WNT signalling in a manner that positively enhances N-hPSC self-renewal[12,23]. The presumptive mechanism of augmented WNT signalling in LIF-3i conditions involves inhibition of tankyrase-mediated degradation of AXIN synergizing with GSK3β-inhibited stabilization of β-catenin. This dual inhibition results in increased AXIN levels which result in stabilization and increased cytoplasmic retention of the activated non-phosphorylated isoform of β-catenin[12,23] (which also decreases nuclear β-catenin-TCF interactions). Importantly, both nuclear (transcriptional) and cytoplasmic (non-transcriptional) activities of activated β-catenin have been suggested to stabilize both naïve and primed pluripotency[12,23]. However, in humans, the repertoire of proteins directly targeted by tankyrase post-translational PARylation extends far beyond modulation of WNT signalling (e.g., AXIN1 and 2, APC2, NKD1, NKD2, and HectD1), and likely includes regulation of a panoply of proteins with diverse biological functions that potentially cooperate to broadly support a stable human naive pluripotent state[47]. These functions include regulation of telomere elongation and cohesion (TRF1), YAP signalling (angiomotin), mitotic spindle integrity (NuMa), GLUT4 vesicle trafficking (IRAP), DNA damage response regulation (CHEK2), and microRNA processing (DICER). Interestingly, TRF1 was identified as an essential factor for iPSC reprogramming in mouse and human PSC[48]. Finally, a critical aspect of the LIF-3i cocktail is its incorporation of MEK inhibition, which is an essential component of classical 2i for maintaining the naïve epiblast state, but was reported to cause aberrant genome-wide DNA hypomethylation in mESC. These epigenetic changes were caused by MEK inhibitor-mediated downregulation of DNA methyltransferases and were irreversible at imprinted loci, and ultimately resulted in the impairment of developmental potential of 2i-treated naive mESC[21]. However, the LIF-3i cocktail did not result in erosion of genomic CpG methylation at imprinted loci in N-hPSC despite inclusion of a MEK inhibitor[12]. Although the mechanism of this imprint preservation by XAV939 despite the presence of a MEK inhibitor currently remains obscure, PARylation has been shown to safeguard the Dnmt1 promoter in mouse cells, and antagonizes aberrant hypomethylation at CpG islands, including at imprinted genes[49,50]. Thus, the role of PARylation on DNA methylation and how it may be modulated by XAV939 requires deeper investigation.

Diabetic hyperglycemic alterations of blood vessel viability and integrity lead to multi-organ dysfunction that results in endothelial dysfunction linked to epigenetic remodeling[17] (e.g., DNA methylation[51], histone marks[52,53], and oxidative stress[54,55]). Several studies have shown that these aberrant epigenetic changes may be partially overcome by genome-wide chemical treatments that restore some endothelial function[56,57]. The extent of retention of diseased diabetic epigenetic memory at developmental genes from incomplete or ineffective reprogramming within DhiPSC-derived lineages and its role in impaired regenerative capacity remains unclear, and marked by high variability in differentiation efficiency or retention of diseased phenotype[58–62]. For example, endothelial differentiation of iPSC generated from

diabetic mice displayed vascular dysfunction, impaired in vivo regenerative capacity, and diabetic iPSC displayed poor teratoma formation[63]. Human iPSC from patients with rare forms of diabetes-related metabolic disorders have similarly shown significant functional endothelial impairment[58]. Transient chemical demethylation of T1D-hiPSC was sufficient to restore differentiation in resistant cell lines and achieve functional differentiation into insulin-producing cells[18].

Naïve reversion of conventional DhiPSC may potentiate an epigenetic remodeling of reprogrammed diabetic fibroblasts that avoids differentiation into dysfunctional ECs with diabetic epigenetic memory. Similarly, tankyrase inhibitor-regulated N-DhiPSC are expected to improve the poor and variable DhiPSC differentiation generation of other affected tissues in diabetes[64] including pancreatic, renal, hematopoietic, retinal, and cardiac lineages. We propose that autologous or cell-banked transplantable progenitors derived from tankyrase inhibitor-regulated N-hiPSC will more effectively reverse the epigenetic pathology that drives diseases such as diabetes. In future clinical studies, multiple cell types (e.g., vascular endothelium, pericytes, retinal neurons, glia, and retinal pigmented epithelium) could all potentially be differentiated from the same autologous or HLA-compatible, banked patient-specific hiPSC line for a comprehensive repair of ischemic vascular and macular degenerative disease. The further optimization of tankyrase/PARP-inhibited human naïve pluripotent stem cells in defined, clinical-grade conditions may significantly advance regenerative medicine.

## Methods

**Bioethics**. hESC lines used in these studies were obtained from the Wisconsin International Stem Cell Bank (WISCB). All hESC experiments conform to guidelines outlined by the National Academy of Sciences, and the International Society of Stem Cell Research (ISSCR). These commercially-acquired hESC were under purview of the Johns Hopkins University (JHU) Institutional Stem Cell Research Oversight (ISCRO), and conformed to Institutional standards regarding informed consent and provenance evaluation. All experiments proposed received approval by the JHU ISCRO committee. All animal use and surgical procedures were performed in accordance with protocols approved by the Johns Hopkins School of Medicine Institute of Animal Care and Use Committee (IACUC) and the Association for Research of Vision and Ophthalmology statement for the Use of Animals in Ophthalmic and Visual Research.

**Conventional primed and naïve hESC and hiPSC cultures**. All hESC and hiPSC lines used in these studies (Supplementary Data 1) were maintained and expanded in undifferentiated conventional feeder-free primed states, or naïve-reverted with the LIF-3i system[12,13]. Conventional cultures of hiPSC were propagated using commercial E8 medium (ThermoFisher Scientific), or an in-house variant formulation consisting of DMEM/F-12 supplemented with 2.5 mM L-Glutamine, 15 mM HEPES and 14 mM sodium bicarbonate (ThermoFisher Scientific, cat# 11330), 50–100 ng/mL recombinant human FGF-basic (Peprotech), 2 ng/mL recombinant human TGF-β1 (Peprotech), 64 μg/mL L-ascorbic acid-2-phosphate magnesium (Sigma), 14 ng/mL sodium selenite (Sigma), 10.7 μg/mL recombinant human transferrin (Sigma), and 20 μg/mL recombinant human insulin (Peprotech). Conventional hiPSC were expanded in E8 onto Vitronectin XF (STEMCELL Technologies) matrix-coated tissue culture-treated 6-well plates (Corning). E8 medium was replaced daily and hiPSC were gently passaged every 5–6 days by mechanical selection or bulk passaged using non-enzymatic reagents (i.e., Versene solution (ThermoFisher Scientific)) or Phosphate-Buffer-Saline (PBS)-based enzyme-free cell dissociation buffer (ThermoFisher Scientific, #13151).

LIF-3i naïve reversion medium was prepared fresh every week and consisted of DMEM/F-12 supplemented with 20% KnockOut Serum Replacement (KOSR, ThermoFisher Scientific), 0.1 mM MEM non-essential amino acids (MEM NEAA, ThermoFisher Scientific), 1mM L-Glutamine (ThermoFisher Scientific), 0.1 mM β-mercaptoethanol (Sigma), 20 ng/mL recombinant human LIF (Peprotech), 3μM CHIR99021 (Tocris or Peprotech), 1 μM PD0325901 (Sigma or Peprotech), and 4 μM XAV939 (Sigma or Peprotech). Prior to switching between E8 and LIF-3i media, hPSC were adapted for one passage in LIF-5i[12,13]. LIF-5i was prepared by supplementing LIF-3i with 10 μM Forskolin (Stemgent or Peprotech), 2μM purmorphamine (Stemgent or Peprotech) and 10 ng/mL recombinant human FGF-basic (Peprotech). Briefly, primed hiPSC were adapted overnight by substituting E8 with LIF-5i medium. The next day, hiPSC were enzymatically dissociated (Accutase, ThermoFisher Scientific) and transferred onto irradiated mouse embryonic fibroblast (MEF) feeders in LIF-5i medium for only one passage (2–3 days). All subsequent passages were grown in LIF-3i medium on MEF feeders.

Isogenic E8 cultures were maintained in parallel for simultaneous phenotypic characterization, as described[12,13].

**Episomal reprogramming of diabetic fibroblasts**. Adult human Type-I diabetic (T1D) donor fibroblasts obtained with patient informed consent, were purchased from DV Biologics, and cultured in fibroblast culture medium (I-Gro medium, DV Biologics). Episomal expression of *SOX2*, *OCT4*, *KLF4*, *c-MYC*, *NANOG*, *LIN28*, *SV40LT* was performed by nucleofection of 1x10⁶ diabetic fibroblast cells with 2 μg each of three plasmids, pCEP4-EO2S-EN2L, pCEP4-EO2S-ET2K, and pCEP4-EO2S-EM2K[27,28]. Single fibroblast cells were obtained with Accutase, and nucleofected using the human dermal fibroblast nucleofector kit (Lonza, VPD-1001) and Amaxa nucleofector program U-023. Nucleofected cells were transferred onto irradiated MEF in fibroblast growth medium supplemented with 10 μM Rho-associated, coiled-coil containing protein kinase (ROCK) inhibitor Y27362 (Stemgent). The next day, 2 mL of DMEM/F-12 supplemented with 20% KOSR, 0.1 mM MEM NEAA, 1 mM L-Glutamine, 0.1 mM β-mercaptoethanol, 50 ng/mL bFGF, 10 μM Y27362, 5 μg/mL ascorbic acid, and 3 μM CHIR99021 was added. Half of the medium was replaced with fresh medium without Y27362 every other day, until hiPSC colonies appeared. Individual hiPSC colonies were manually isolated, expanded onto vitronectin-coated plates in E8 medium, or further expanded and cryopreserved.

**Isogenic primed vs. naïve hiPSC directed differentiation**. To examine the differentiation performance of normal and diabetic N-hiPSC, we directly differentiated LIF-3i-reverted naïve vs. their primed genotypically-identical isogenic sibling hiPSC counterparts in parallel, without additional cell culture manipulations[12,13]. Re-priming (i.e., converting N-hiPSC back to conventional primed conditions prior to their use in directed differentiation assays[25,26]) was not necessary with the LIF-3i method[12,13]. To minimize variations within directed differentiation experiments that may arise from hiPSC interline variability and genetic background, paired isogenic primed and LIF-3i-reverted hiPSC lines were simultaneously and cultured into defined, identical, feeder-free differentiation systems according to manufacturer's directions. Naïve reversions were performed in LIF-5i/LIF-3i media fresh for each differentiation experiment starting from a low passage primed hPSC line[13]. Additionally, functional comparisons of naïve vs. primed isogenic hiPSC lines, sibling cultures were prepared at equivalent passage number, starting from the primed parental hPSC line. Primed and naïve hPSC sibling cultures were expanded in parallel in their respective media for 5–7 passages before differentiation (e.g., E8 vs. LIF-3i, see schematic Fig. 3b). Detailed information for the origins and derivation of all hiPSC lines used in these studies for these assays was previously published[12] (Supplementary Data 1).

Neural differentiation was performed using GIBCO PSC neural induction medium (NIM; ThermoFisher Scientific, A1647801) and the manufacturer's recommendations. Differentiation into definitive endodermal progenitors was performed using the StemDiff Definitive Endoderm Kit (StemCell Technologies) following manufacturer's protocols. The experimental approach for vascular differentiation of VP from primed and naïve hiPSC was modified and optimized from methods described[7], and is summarized in Supplementary Fig. 3. Briefly, VP differentiation was performed using a modification of the STEMdiff APEL-Li medium system[32]. APEL-2Li medium (StemCell Technologies, #5271) was supplemented with Activin A (25 ng/mL), VEGF (50 ng/mL), BMP4 (30 ng/mL), and CHIR99021 (1.5 μM) for the first 2 days, and then APEL-Li that was supplemented with VEGF (50 ng/mL) and SB431542 (10 μM). Differentiation medium was replaced every 2 days until cells were harvested for analysis.

**Isogenic hiPSC teratoma assays**. Isogenic primed (E8) and naïve (LIF-3i) hiPSC cultures were maintained in parallel for 9 passages prior to teratoma formation assays. Teratomas from the same hiPSC line were directly generated from a fixed number of cells (5x10⁶) and duration (8 weeks) in primed vs. LIF-3i naïve conditions. LIF-3i-cultured N-hiPSC colonies did not require additional chemical manipulation or re-priming culture steps prior to enzymatic harvest from culture and direct injection of cells into immunodeficient NOG mice. Adherent primed vs. naïve hiPSC were collected using Accutase and counted using Countess counter (ThermoFisher Scientific). For all experiments, 5 × 10⁶ hiPSC were admixed with Growth factor reduced Matrigel (Corning, cat# 356230) on ice. Cells were injected subcutaneously into the hind limbs of immunodeficient NOG male sibling mice. Teratomas were dissected 8 weeks following injection and fixed by overnight immersion in PBS, 4% formaldehyde. All tissues were paraffin-embedded, and microsectioned (5 μm thickness) onto microscope glass slides (Cardinal Health) by the Histology laboratory from the Pathology Department at the Johns Hopkins University. To account for heterogeneous teratoma histological distribution, 15 individual equally spaced sections were immunostained per tissue for each antigen of interest and quantification. Slides were heated in a hybridization oven (ThermoFisher Scientific) at 60 °C for 20 min and then kept at room temperature for 1 h to dry. Paraffin was eliminated by three consecutive immersions in xylenes (Sigma) and sections were rehydrated by transitioning the slides in successive 100%, 95%, 70% and 0% ethanol baths. Sections were placed in 1X wash buffer (Dako) prior to heat-induced antigen retrieval using 1X Tris-EDTA, pH9 target retrieval solution (Dako) and wet autoclave (125 °C, 20 min). Slides were cooled and progressively

transitioned to PBS. After 2 washes, tissues were blocked for 1 h at room temperature using PBS, 5% goat serum (Sigma), 0.05% Tween 20. Endogenous biotin receptors and streptavidin binding sites were saturated using the Streptavidin/Biotin Blocking kit (Vector Laboratories). All antibodies were diluted in blocking solution. Sections were incubated overnight at 4 °C with monoclonal mouse anti-NG2 (Sigma, C8035, 1:100), mouse anti-SOX2 (ThermoFisher Scientific, MAS-15734, 1:100) or rabbit anti-cytokeratin 8 (Abcam, ab53280, 1:400) primary antibodies, washed 3 times, incubated for 1 h at room temperature with biotinylated goat anti-mouse or goat anti-rabbit IgG antibodies (Dako, 1:500), washed three times and incubated with streptavidin Cy3 (Sigma, 1:500) for 30 min at room temperature. After two washes, tissues were incubated for 2 h at room temperature with a second primary antibody (e.g., anti-Ki-67) differing in species from the first primary antibody. After incubation with rabbit (Abcam, ab16667, 1:50) or mouse (Dako, M7240, 1:50) anti-Ki-67 monoclonal antibody, sections were washed three times and incubated for 1 h at room temperature with highly cross-adsorbed Alexa Fluor 488-conjugated goat anti-rabbit or goat anti-mouse secondary antibody (ThermoFisher Scientific, 1:250). Sections were washed twice, incubated with 10 µg/mL DAPI (ThermoFisher Scientific, D1306) in PBS, washed three times in PBS and slides were mounted with coverslips using Prolong Gold Anti-fade reagent (ThermoFisher Scientific) for imaging. Isotype controls for mouse (ThermoFisher Scientific) and rabbit (Dako) antibodies were substituted at matching concentration with primary antibodies as negative controls.

For teratoma organoid quantifications, photomicrographs were obtained using a ×20 objective and Zeiss LSM 510 Meta Confocal Microscope. Teratoma organoid quantifications were first assessed by histological grading of 20 whole cross-sections that were equally spaced throughout the tissue and stained with hematoxylin-eosin. Lineage-specific quantifications were validated in adjacent sections ($n = 15$) by fluorescent immunostains. Image processing and quantification was performed using NIS-Elements software (Nikon). The ROI editor component was applied to autodetect regions of interest in the Cy3 channel that delineated lineage-defined structures (i.e., Cytokeratin 8+ definitive endoderm, NG2+ chondroblasts, SOX2+ neural rosettes) within teratomas. Thresholding and restrictions were standardized in the Object Count component and applied to detect and export the number of DAPI+ and Ki67+ nuclei within ROIs for all analyzed sections.

**Antibodies.** Source and working dilutions of all antibodies used in these studies for Western blots, FACS, genomic dot-blots, ChIP, and immunofluorescence experiments are listed in Supplementary Data 3.

**Western blotting.** Cells were collected from either primed (E8 medium) on vitronectin-coated plates or naïve (LIF-3i/MEF plates) conditions with Enzyme-Free Cell Dissociation Buffer (Gibco, 13151-014). Cells were washed in PBS and pelleted. Cell pellets were lysed in 1× RIPA buffer (ThermoFisher Scientific, 89900), 1mM EDTA, 1× Protease Inhibitor (ThermoFisher Scientific, 78430), and quantified using the Pierce bicinchoninic acid (BCA) assay method (ThermoFisher Scientific). 25 µg of protein per sample was loaded on a 4–12% NuPage Gel (ThermoFisher Scientific, NP0336) according to manufacturer's recommendations. The gel was transferred using the iBlot2 (Life Technologies), blocked in Tris-buffered saline (TBS), 5% nonfat dry milk (Labscientific), 0.1% Tween-20 (TBS-T) for 1 h, and incubated overnight at 4 °C in with anti-phosphorylated-STAT3 primary antibody (Cell Signaling, 9145) according to manufacturer's protocols. Membranes were rinsed three times in TBS-T, incubated with horseradish peroxidase (HRP) –linked goat anti-rabbit secondary antibody (Cell Signaling, 7074) for 1 h at room temperature, rinsed three times, and developed using Pierce ECL Substrate (ThermoFisher Scientific, 32106). Chemiluminescence detection was imaged using an Amersham Imager 600 (Amersham). Anti-actin antibody staining was performed for each membrane as a loading control. Quantitative densitometry was performed on all Western blot images presented in this study using ImageJ software. Semi-quantitative densitometry was measured using the ImageJ software and normalized to actin controls.

**Flow cytometry of vascular differentiation and purification of endothelial-pericytic VP.** Recipes for all differentiation reagents, antibodies, and PCR primers are described, and summarized in Supplementary Data 3. For flow cytometry analysis of vascular differentiations, cells were washed once in PBS, and enzymatically digested with 0.05% trypsin-EDTA (5 min, 37 °C), neutralized with FCS, and cell suspensions were filtered through a 40 µm cell-strainer (Fisher Scientific, Pittsburgh, PA). Cells were centrifuged ($200 \times g$, 5 min, room temperature) and resuspended in staining buffer (EBM alone or 1:1 EMG2:PBS). Single cell suspensions (<1x10$^6$ cells in 100 µL per tube) were incubated for 20 min on ice with directly conjugated mouse monoclonal anti-human antibodies and isotype controls. Cells were washed with 3 mL of PBS, centrifuged ($300 \times g$, 5 min, room temperature), and resuspended in 300 µL of staining buffer prior to acquisition. Viable cells were analyzed (10,000 events acquired for each sample) using the BD CellQuest Pro analytical software and FACSCalibur™ flow cytometer (BD Biosciences). All data files were analyzed using Flowjo analysis software (Tree Star Inc., Ashland, OR).

FACS of primed vs. naïve VP populations was performed at the Johns Hopkins FACS Core Facility with a FACS Aria III instrument (BD Biosciences, San Jose,

CA). Cell suspensions from APEL vascular differentiations were incubated with mouse anti-human CD31-APC (eBioscience, San Diego, CA) and CD146-PE (BD Biosciences) antibodies for 30 min on ice, and FACS-purified for high CD31 and CD146 expression, plated onto fibronectin-coated plates in EGM2, and expanded to 80–90% confluency for 7–9 days prior to in vitro analyses or in vivo injections into the eyes of I/R-treated NOG mice.

**Vascular functional assays.** The methods for endothelial Dil-acetylated-LDL uptake assays, Matrigel tube quantitation assays, EdU proliferation assays, β-galactosidase senescent assays were described[7], and are summarized briefly below. For Dil-Acetylated-Low Density Lipoprotein (Dil-Ac-LDL) uptake assays, FACS-purified CD31+CD146+ primed vs. naïve VP populations were expanded in EGM2 medium ~7 days to 60–70% confluency on fibronectin pre-coated 6-well plates ($1–1.5 \times 10^5$ cells/well) prior to Dil-Ac-LDL uptake assays (Life Technologies, Cat No. L-3484). Fresh EGM2 medium supplemented with 10 µg/mL Dil-Ac-LDL, was switched before assays, and incubated for 4 h at 37 °C. Cells were washed in PBS and Dil-Ac-LDL-positive cells imaged with a Nikon Eclipse Ti-u inverted microscope (Nikon Instruments Inc., Melville, NY) and NIS Elements imaging software. Cells were also harvested with Accutase (5 min, 37 °C) and Dil-Ac-LDL+ cells quantitated by flow cytometry.

In vitro vascular functionality of primed DVP vs. N-DVP was determined with quantitative Matrigel vascular tube-forming assays[7]. Briefly, MACS-purified CD31+ CD146+ isogenic DVP were expanded in EGM2 on fibronectin-coated (10 µg/mL) tissue culture plates. Adherent cells were treated with Accutase for 5 min, and collected into single cell suspensions. Primed DVP or N-DVP cells were transferred into 48-well plates ($2 \times 10^5$ cells/ well in EGM2 medium) pre-coated with Matrigel (Corning, #356237, 200 µL/well). The next day, multiple phase contrast pictures of vascular tube formations were imaged with an inverted Eclipse Ti-u Nikon microscope (Nikon Instruments Inc., Melville, NY) and NIS Elements imaging software without overlapping the imaged regions. All the vascular tubes formed by VP, DVP, and N-DVP were measured by NIS-Elements imaging software. Statistical comparisons were performed with unpaired t-tests using Prism (GraphPad Software, San Diego, CA).

For senescence assays, naïve vs. primed VP populations were plated onto fibronectin (10 µg/mL)-coated 6-well tissue culture plates, and VP were expanded in EGM2 for up to 30 days (3–6 passages), and senescent cells were assayed for acidic senescence-associated ß-galactosidase activity. Cells were grown to ~60–80% confluency in 12-well fibronectin-coated plates prior to analysis. Cells were fixed in 2% paraformaldehyde and β-galactosidase activity was quantified by detecting hydrolysis of the X-gal substrate by colorimetric assay as per manufacturer's protocol for detection of senescent cells. (Cell Signaling Technology, Danvers, MA). Nuclei were counterstained using the fluorescent dye Hoechst 33342 (BD Biosciences). Total number of Hoechst+ cells and blue X-Gal+ senescent cells were automatically enumerated using an inverted Eclipse Ti-u Nikon microscope and the Object Count component of the NIS Elements software. For each sample, 2 individual wells were photographed at five independent locations using a ×20 objective.

**Transmission electron microscopy (TEM).** Primed vs. naïve VP were plated onto fibronectin (10 µg/mL) coated Labtek chambers, culture expanded in EGM2, and fixed for TEM at the Wilmer Microscopy Core[7]. Sections were imaged with a Hitachi H7600 TEM at 80KV (Gaithersburg, MD) and a side mount AMT CCD camera (Woburn, Mass).

**NCS DNA damage response assays.** Primed DhiPSC and N-DhiPSC isogenic (same lines at same passage) were simultaneously differentiated in parallel into DVP and N-DVP using APEL medium, as above. CD31+CD146+ VP cells were expanded in EGM2 (3 passages) onto fibronectin (10 µg/mL) 6-well plates (for Western blot analysis), or alternatively the last passage was transferred onto 8-well Nunc Labtek II chamber slides for immunostaining. To induce DNA damage, expanded DVP and N-DVP cells were incubated for 5 h in EGM2 supplemented with 100 ng/mL of the radiomimetic agent neocazinostatin (NCS, Sigma). Untreated DVP and N-DVP cells were analyzed in parallel as controls. Western blot analysis was performed as described above. For detection of phosphorylated H2AX by immunofluorescence, VP cells were fixed for 10 min using 1% paraformaldehyde in PBS. For immunofluorescent staining of chambered slides, fixed cells were blocked for non-specific staining and permeabilized using a blocking solution consisting of PBS, 5% goat serum (Sigma) and 0.05% Tween 20 (Sigma). Samples were incubated overnight at 4 °C with a rabbit anti-human phospho-H2AX antibody (Cell Signaling, #9718) diluted (1:200) in blocking solution. The next day, VP cells were washed (Dako wash buffer, Dako) and incubated for 2 h at room temperature with a biotinylated goat anti-rabbit secondary antibody (Dako, 1:500 in blocking solution). Cells were washed three times and incubated for 30 min with streptavidin Cy3 (Sigma, 1:500). All samples were sequentially washed and incubated with a mouse monoclonal anti-human CD31 (Dako, M0823, 1:100) and Alexa488-conjugated goat anti-mouse secondary antibody (ThermoFisher, 1:100), both for 1 h at room temperature. Finally, slides were washed in PBS and incubated with DAPI (1:2000) for 5 min at room temperature for nuclear staining. Slides were mounted using the Prolong Gold anti-fade mounting reagent (ThermoFisher) and cured overnight. For each condition, 5–6 independent frames were

captured for the Cy3, Alexa488 and DAPI channels using a ×20 objective and a LSM510 Meta confocal microscope (Carl Zeiss Inc., Thornwood, NY) in the Wilmer Eye Institute Imaging Core Facility. Quantification of phospho-H2AX+ foci within DAPI+ nuclei of CD31+ VP was performed using the NIS-Elements software. Briefly, thresholds and masks were sequentially created for the Alexa488 (CD31) and DAPI channels to limit the analysis to nuclei of VP cells. Nuclei were further defined using the size/area and circularity parameters. Each individual CD31+ nucleus was characterized as a single object using the object count function. Finally, the number of foci per nucleus was determined by counting the number of objects in the Cy3 channel. A total of 128–165 nuclei were analyzed for each condition (primed vs. naïve ± NCS). Statistical comparisons of the distribution of number of phospho-H2Ax+ foci per nuclei between VP populations were assessed by Chi-square test (z-test) using Graphpad Prism.

**Ocular I/R Injury and VP Injections into NOD/Shi-scid/IL-2Rγ^null^ (NOG) eyes.** The I/R ocular injury model was previously published[7]. Briefly, six- to eight-week-old male NOG mice (Johns Hopkins Cancer Center Animal Facility) were subjected to high intraocular pressure to induce retinal ischemia-reperfusion injury. Mice were deep anesthetized by intraperitoneal (IP) injection of ketamine/xylazine (50 mg/kg ketamine + 10 mg/kg xylazine in 0.9% NaCl). The pupils were dilated with 2.5% phenylephrine hydrochloride ophthalmic solution (AK-DILATE, Akorn, Buffalo Glove, IL) followed by 0.5% tetracaine hydrochloride ophthalmic topical anesthetic solution (Phoenix Pharmaceutical, St. Joseph, MO). The anterior chamber of the eye was cannulated under microscopic guidance (OPMI VISU 200 surgical microscope, Zeiss, Gottingen, Germany) with a 30-gauge needle connected to a silicone infusion line providing balanced salt solution (Alcon Laboratories, Fort Worth, TX); avoiding injury to the corneal endothelium, iris, and lens. Retinal ischemia was induced by raising intraocular pressure of cannulated eyes to 120 mmHg for 90 min by elevating the saline reservoir. Ischemia was confirmed by iris whitening and loss of retinal red reflex. Anesthesia was maintained with two doses of 50 μL intramuscular ketamine (20 mg/mL) for up to 90 min. The needle was subsequently withdrawn, intraocular pressure normalized, and reperfusion of the retinal vessels confirmed by reappearance of the red reflex. The contralateral eye of each animal served as a non-ischemic control. Antibiotic ointment (Bacitracin zinc and Polymyxin B sulfate, AK-Poly-Bac, Akron) was applied topically. 2 days later, MACS-purified and expanded human DVP and N-DVP were injected into the vitreous body (50,000 cells in 2 μL/eye), using a micro-injector (PLI-100, Harvard Apparatus, Holliston, MA).

**Immunofluorescence (IF) staining of whole-mounted mouse retinae.** Human cell engraftment into NOG mouse retinae were detected directly with anti-human nuclear antigen (HNA) immunohistochemistry with murine vascular marker co-localization (murine CD31 and collagen IV) using anti-murine CD31 and anti-murine collagen IV antibodies. Animals were euthanized for retinal harvests and HNA-positive cell quantitation at 1, 3, and 4 weeks following human VP injection (2 days post-I/R injury). After euthanasia, eyes were enucleated, cornea and lens were removed, and the retina was carefully separated from the choroid and sclera. Retinae were fixed in 2% paraformaldehyde in TBS for overnight at 4 °C, and permeabilized via incubation with 0.1% Triton-X-100 in TBS solution for 15 min at 4 °C. Following thorough TBS washes, free floating retinas were blocked with 2% normal goat serum in TBS with 1% bovine serum albumin and incubated overnight at 4 °C in primary antibody solutions: rabbit anti-mouse Collagen IV (AB756P, Millipore, 1:100) and/or rat anti-mouse CD31 (550274, BioSciences, 1:50) in 0.1% Triton-X-100 in TBS solution (to label basement membrane and EC of blood vessels, respectively). On the next day, retinae were washed with TBS, and incubated with secondary antibodies for 6 h at 4 °C. A goat anti-rabbit Cy3-conjugated secondary antibody (Jackson Immuno Research, # 111-165-003, 1:200) was used to detect collagen IV primary antibody, and a goat anti-rat Alexafluor-647-conjugated secondary antibody (Invitrogen, # A21247, 1:200) was used to detect the anti-CD31 primary antibody. Human cells were detected using directly Cy3-conjugated anti-HNA (Millipore, MAB1281C3, 1:100). After washing in TBS, flat mount retinas were imaged with confocal microscopy (LSM510 Meta, Carl Zeiss Inc., Thornwood, NY) at the Wilmer Eye Institute Imaging Core Facility.

**Confocal microscopy and quantitation of human cell engraftment in murine retinae.** For quantification of HNA+ cells in the superficial layers of whole retinae, whole mount retinas were prepared from the eyes of animals at 1, 3 or 4 weeks following intra-vitreal transplantation of human cells (50,000 primed DVP or N-DVP cells per eye) following I/R injury. Non-I/R-injured eyes and control PBS-injected eyes were also analyzed as controls. Images were acquired with ZEN software using a ×10 objective and a LSM510 Meta confocal microscope. For each individual eye, the entire retina was tile-scanned and stitched (7x7 frames, 10% overlapping).

For human HNA+ cell quantification analysis, photomicrographs were processed using the Fiji distribution of imageJ. Briefly, a region of interest was created using the DAPI channel and the 'magic wand' function to conservatively delineate the whole retina and exclude from the analysis the limited background at the edges of the retina preparation that could be detected in the Cy3 (HNA) channel for some samples. The Cy3 channel was processed with the 'smooth' function and a mask was created using the thresholding function. The Cy3 channel

was further prepared for the 'Analyze particle' plugin by using standardized sequential corrections that were limited to despeckle, filtering (Minimum) and watersheding. Particle objects corresponding to HNA+ nuclei were automatically counted using fixed size and circularity parameters.

Eyes were also analyzed for quantification of human CD34+ or human CD31+ blood vessels within defined layers of the mouse retina in some experiments. Briefly, the anterior eye (cornea/iris) was dissected free by a circumferential cut at the limbus. Eyecups were fixed using paraformaldehyde and prepared for cryopreservation by immersion in gradients of sucrose. Eyes were hemisected through the optic nerve (Supplementary Fig. 8a) and the two halves embedded in OCT-sucrose. Serial cryosections (8 μm thickness) were prepared from hemisections that included the retina (Supplementary Fig. 8a), and stored at −80 °C. Equally interspaced microsections [n = 11 (E8) and 13 (LIF-3i) for CD34, and n = 3 (E8) and 7 (LIF-3i) for CD31 immunostainings]. Retinal sections were sequentially immunostained with either mouse anti-human CD34 (BD Biosciences, clone My10, #347660, 1:50) or mouse anti-human CD31 (Dako, M0823, 1:50) overnight at 4 °C followed by goat anti-mouse Alexa488 (ThermoFisher Scientific, 1:200) for 1 h at room temperature. Sections were subsequently stained with either rabbit anti-mouse collagen type IV (Millipore, AB756P, 1:200) or rat anti-mouse CD31 (BD Biosciences, 550274, 1:50) for 1 h at room temperature. Alexa647-conjugated goat anti-rabbit (ThermoFisher Scientific, A-21246; 1:200) or goat anti-rat (ThermoFisher Scientific) secondary antibodies (i.e., conjugated F(ab')2 fragments) that were highly cross-adsorbed against IgG from other species were subsequently incubated for 1 h at room temperature. Nuclei were counterstained using DAPI. Negative immunostaining controls in each experiment were conducted and confirmed negative, and consisted of replacing primary antibodies with mouse, rat and rabbit nonimmune IgG (Dako or ThermoFisher Scientific) at the corresponding antibody concentration to verify absence of unspecific antibody binding. Retinal sections were mounted with Prolong Gold anti-fade reagent (ThermoFisher Scientific) and cured overnight in the dark. Images were acquired using a ×20 objective with the ZEN software and a LSM510 Meta confocal microscope.

Photomicrographs were further processed for human cell quantification using the Fiji distribution of imageJ (Supplementary Fig. 8b). Briefly, regions of interest (ROI) were created using the DAPI channel as a template to delineate the GCL, INL, and ONL, the other regions (ILM, IPL, OPL, and S) being defined as intercalated around and between the 3 DAPI-defined ROI. Analysis was pursued by processing the Alexa488 (human CD34 or CD31) channel using a sequential series of defined parameters using the 'smooth', 'despeckle', 'filter (median)' functions and thresholding. Alexa488+ objects were counted within and between ROI using ImageJ. The numbers of human CD34+ or CD31+ blood vessels were automatically enumerated using the 'Analyze particles' plugin within Fiji. Images were captured using a 20X objective (450 μm $^2$). Speckles and non-specific background (<10 pixels) were excluded. Regions delineating the retinal layers were created using the DAPI channel. A smoothened image was segmented by thresholding the CD34 or CD31 signal (pixel values >50). The ImageJ module 'Analyze particles' was set to select object with a surface area >5 μm$^2$ and each image automatic count was cross-validated by manual counting of threshold images and exclusion of duplicate objects (2 or more objects belonging to blood vessels that were longitudinally cross-sectioned). Human CD34 (Supplementary Fig. 8b) or human CD31 expression was scored only when the Alexa488+ signal was expressed at chimeric human-murine blood vessels that also expressed either murine collagen IV (mColIV) or murine CD31 (mCD31) (Fig. 7a,c). The quantity of human blood vessels detected in murine vessels ranged between 1–13 per image at ×20 objectives (Fig. 7b–d).

**Quantitative real-time polymerase chain reaction (qRT-PCR) and chromatin immunoprecipitation PCR (ChIP-qPCR).** The sequences and published reference citations of all PCR primers used in these studies for qRT-PCR and qChIP-PCR are catalogued in Supplementary Data 3. For qRT-PCR analyses, feeder-dependent LIF-3i hPSC cultures were MEF-depleted by pre-plating onto 0.1% gelatin-coated plates for 1 h at 37 °C [12]. Samples were sequentially and simultaneously collected from representative hPSC lines in primed (E8), or naïve (LIF-3i; p > 3) conditions. Alternatively, genotypic-identical (isogenic) paired samples were prepared from EGM2-expanded primed and naïve VP. Total RNA was isolated from snap-frozen samples using the RNeasy Mini Kit (Qiagen) following the manufacturer's instructions, and quantified using a Nanodrop spectrophotometer (ThermoFisher Scientific). Genomic DNA was eliminated by in-column DNase (Qiagen) digestion. Reverse transcription of RNA (1μg/sample) was accomplished using the Super-Script VILO cDNA Synthesis Kit (ThermoFisher Scientific) and a MasterCycler EPgradient (Eppendorf). For real-time qPCR amplification, diluted (1:20) cDNA samples were admixed to the TaqMan Fast Advanced Master Mix (ThermoFisher Scientific) and Taqman gene expression assays (ThermoFisher Scientific).

Matching isogenic samples were prepared in parallel for ChIP-qPCR assays. Isogenic hPSC cultures were expanded using primed (E8) and naïve (LIF-3i/MEF) conditions and analyzed at passages matching qRT-PCR analysis. Alternatively, VP cells were prepared from isogenic primed and naïve PSC using the same APEL/EGM2 conditions as the samples prepared for RT-PCR. Cells were collected using Accutase and counted using a Countess cell counter (ThermoFisher Scientific). Feeders were excluded from LIF-3i/MEF samples by pre-plating for 1 h on gelatin-coated plates and preplated samples were re-counted after the pre-plating step. 3 ×

$10^6$ cells were allocated per ChIP assay and prepared using the Magna ChIP A/G chromatin immunoprecipitation kit (Millipore). Cells were centrifuged (300 g), supernatant was discarded and cells were fixed for 10 min at room temperature by resuspending in 1mL of PBS, 1% formaldehyde (Affymetrix). Unreacted formaldehyde was quenched using 100 μL 10× Glycine (Millipore). Samples were left at room temperature for 5 min, centrifuged (300g) and washed twice in 1mL ice-cold PBS. Samples were resuspended in ice-cold PBS containing either 1× Protease Inhibitor Cocktail II (Millipore) or 1X complete Mini protease inhibitor (Roche). Samples were centrifuged at $800 \times g$ for 5 min, cell pellets were snap-frozen in liquid nitrogen and stored at $-80\,°C$ until use for ChIP assay. Cell lysis, homogenization and nuclear extraction of cryopreserved samples were processed using the reagents provided in the Magna ChIP kit and the manufacturer instructions. The isolated chromatin was fragmented using a Diagenode Bioruptor Plus sonication device. Sonication settings (10 cycles, high, 30 s on, 30 s off) were validated in pilot experiments to shear cross-linked DNA to 200–1000 base pairs by agarose gel electrophoresis. The sheared chromatin was centrifuged at $10,000 \times g$ at $4\,°C$ for 10 min and immediately processed for immunoprecipitation. $1 \times 10^6$ cell equivalent of cross-linked sheared chromatin were prepared according to the kit manufacturer's protocol. Briefly, 1% of sheared chromatin was separated as 'input' control. The remaining sample was admixed with 5 μg of immuno-precipitating antibody (Supplementary Data 3) and protein A/G magnetic beads. Antibodies were substituted with corresponding rabbit or mouse IgG (Supplementary Data 3) as negative isotype controls using 5% sheared chromatin. The chromatin-antibody-beads mixture was left incubating overnight at $4\,°C$ with agitation. Protein A/G beads were pelleted using a MagJET separation rack (ThermoFisher Scientific) and supernatant was discarded. Protein/DNA complexes were washed and eluted, beads were separated using the MagJET rack and DNA was purified according to the manufacturer's instructions. The immunoprecipitated genomic DNA was amplified using the Power SYBR Green Master Mix (ThermoFisher Scientific) with relevant published primers (Supplementary Data 3) for GAPDH, GATA2, GATA6, HAND1, NANOG, MSX2, PAX6, SOX1, CD31, vWF, endothelin-1, ICAM2, MYOD1, CXCR4, DLL1, FZD7 and ELP3[65] using a ViAA7 Real Time PCR System (ThermoFisher Scientific). Specificity of antibodies was validated using the isotype controls and samples were normalized to their corresponding input controls.

**Genomic DNA dot-blots of 5-methylcytosine (5MC) and 5-hydroxymethyl-cytosine (5hMC) CpG methylation**. Genomic DNA from isogenic, parallel primed (E8) and preplated naïve (LIF-3i) cultures of representative hiPSC lines was extracted using the DNeasy Blood and tissue Kit (Qiagen) and quantified using a Nanodrop spectrophotometer (ThermoFisher Scientific)[12]. For each sample, 1.6 μg DNA was diluted in 50μL of nuclease-free water (Ambion), denatured by adding 50μL of 0.2M NaOH, 20 mM EDTA and incubating for 10 min at 95 °C, and neutralized by adding 100 μL 20X Saline-Sodium Citrate SSC hybridization buffer (G Biosciences) and chilling on ice. A series of five 2-fold dilutions (800–50 ng) and nuclease-free water controls were spotted on a pre-wetted (10X SSC buffer) nylon membrane using a Bio-Dot Microfiltration Apparatus (Bio-Rad). The blotted membrane was air-dried and UV-cross-linked at 1200 $J/m^2$ using a UV Stratalinker 1800 (Stratagene). The membrane was blocked in TBST, 5% nonfat dry milk for 1 h at room temperature with gentle agitation, washed three times in TBST, and incubated at $4\,°C$ overnight with rabbit anti-5mC (Cell Signaling, 1:1000) or anti-5hmC (Active Motif, 1:5000) antibodies diluted in TBST, 5% BSA. The membrane was washed three times in TBST and incubated for 1 h at room temperature with HRP-conjugated anti-rabbit secondary antibody (Cell Signaling) diluted 1:1000 in blocking buffer. After three washes in TBST, the membrane was treated with Pierce ECL Substrate (ThermoFisher) for chemiluminescent detection with an Amersham Imager 600 (Amersham). After acquisition, the membrane was washed three times in $H_2O$ and immersed in 0.1% methylene blue (Sigma), 0.1M sodium acetate stain solution for 10 min at room temperature. Excess methylene blue was washed three times in water with gentle agitation. Colorimetric detection was performed using the Amersham Imager 600 (Amersham). 5mC, 5hMC, and methylene blue densitometric intensities were quantified by ImageJ software.

**Statistics**. Statistical significance was determined using statistical graphing software (Prism GraphPad) using two-tailed $t$-tests (between individual groups), or 1-way analysis of variance (e.g., analysis of variance-Eisenhart method with Bonferroni correction) for statistical testing of $\geq 3$ groups. For smaller, non-Gaussian–distributed sample sizes ($n < 10$), nonparametric (Mann-Whitney) tests were performed. $P$-values $< 0.05$ were considered significant. All error bars of data displayed denote the standard error of the mean (SEM).

**RNA-Seq, CpG methylation arrays, and bioinformatics analyses**. For RNA-Seq studies, strand specific mRNA libraries were generated using the NEBNext Ultra II Directional RNA library prep Kit for Illumina (New England BioLabs #E7760); mRNA was isolated using Poly(A) mRNA magnetic isolation module (New England BioLabs #E7490). Preparation of libraries followed the manufacturer's protocol (Version 2.2 05/19). Input was 1μg and samples were fragmented for 15 min for RNA insert size of ~200 bp. The following PCR cycling conditions were used: 98 °C 30s / 8 cycles: 98 °C 10s, 65 °C 75s / 65 °C 5 min. Stranded mRNA libraries were sequenced on an Illumina HiSeq4000 instrument using 47bp paired-end dual indexed reads and 1% of PhiX control. mRNA sequencing depth ranged from 30–100M reads. Reads

were aligned to GRCh38 using STAR version 2.7.2b[66] with the following options–readFilesCommand zcat–outSAMtype BAM Unsorted SortedByCoordinate–quantMode TranscriptomeSAM GeneCounts –outFileNamePrefix. We generated summarized experiment objects using the gtf file Homo_sapiens.GRCh38.97.gtf and the following command from the Bioconductor package 'GenomicAlignments': summarizeOverlaps (features=exonsByGene, reads=bamfiles, mode="Union", singleEnd=FALSE, ignore.strand=FALSE, fragments=TRUE. Differential expression analysis and statistical testing was performed using DESeq2 software[67].

The gene expression microarrays (Illumina Human HT-12 Expression BeadChip, San Diego, CA) and Infinium 450K CpG methylation raw array data analyzed in these studies were published previously and available at Gene Expression Omnibus under accession numbers GSE65211 and GSE65214, respectively[12]. Gene-specific enrichment analysis (GSEA) of expression arrays was conducted as described[68,69]. The bioinformatics method for calculating cross-plots of differential promoter CpG methylation beta values vs. corresponding differential gene expression was previously described[12].

**Reporting summary**. Further information on research design is available in the Nature Research Reporting Summary linked to this article.

## Data availability
The source of all quantitative graphs (Figs. 1b–d, 2d, 3a–e, 4b, c, 5b, 6c–f, 7b–d, Supplementary Figs. 1a,c, 3a, and 5c, and all raw Western. blot images, densitometry quantitations, and FACS plots presented in main and supplementary figures are compiled in a file folder entitled Source Data and available online as Supplemental Information. The NIH Gene Expression Omnibus has issued the accession number GSE141639 for RNA-Seq data in this manuscript.

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

## Acknowledgements

This work was supported by grants from the NIH/NEI (R01EY023962), NIH/NICHD (R01HD082098), Novo-Nordisk Diabetes & Obesity Science Forum Award, RPB Stein Innovation Award, The Maryland Stem Cell Research Fund (2018-MSCRFV-4048, 2014-MSCRFE-118153), an RPB Unrestricted grant (Wilmer), The Lisa Dean Moseley Foundation, and Wilmer core grant for vision research (EY001765). We are grateful for technical support by Schuyler Metzger, Jessica Davidson, Rakel Tryggvadottir, and Adrian Idrizi.

## Author contributions

T.S.P., L.Z., and E.T.Z. designed all experiments and wrote the manuscript. T.S.P., L.Z, R.E.M., J.T., J.S.H., R.K., A.H., N.R., R.G., I.B., M.B., M.A.K., and G.L. performed and/or analyzed experiments. All authors edited the manuscript, interpreted results, and gave final approval of the manuscript. E.T.Z. supervised and directed the studies, and wrote the final manuscript.

## Competing interests

The authors declare no competing interests.
