## [Peer Review File · Nature Communications]

Reviewers' comments:

Reviewer #1 (Remarks to the Author):

Vascular Progenitors Generated from Tankyrase Inhibitor-Regulated Diabetic Induced Pluripotent Stem Cells Potentiate Efficient Revascularization of Ischemic Retina

The authors present a novel and highly relevant manuscript, in which they utilize naïve human induced pluripotent stem cells (N-iPSCs) from diabetic donors to facilitate recovery of vascular injury in an ischemic model of retinopathy. This is relevant to diabetic retinopathy, as vascular damage is a primary disease process. The use of reprogrammed fibroblasts to repair vascular damage has therapeutic potential for diabetic retinopathy, and potentially for other diabetic complications and retinopathies of other etiologies.

The authors convincingly demonstrate that N-iPSCs have a less-differentiated phenotype than conventionally prepared iPSCs. Their findings also suggest that iPSCs reprogrammed with this method have a more reparative phenotype, even when the cells are derived from diabetic donors, as would be the case in the context of diabetic retinopathy.

The manuscript has a few weaknesses that should be addressed. In its current form, the manuscript is very difficult to follow due to the extensive use of abbreviations, and inconsistent use of terminologies. Please re-read the manuscript carefully and ensure that all terminologies are consistent. Some of the models used are not described or justified, as specified below. Finally, some of the results appear to lack requisite statistical analyses and essential controls, as specified in the figure critique and recommended additional experiments.

Figure Critique and Recommended Additional Experiments

Figure 1

- The use of "E8" is confusing, as this is not explained in the results section. Please either explain that this signifies E8 media, or label the figure differently, for example with "control."
-

Figure 2

- Where possible a quantitative assessment of images should be included in the main figures.

Figure 3

- 3b-d: Please show statistics
- 3b-c: Please include legends for graphs
- 3d: Please explain the rationale for comparing to CB-iPSC. What is the efficiency in diabetic versus non-diabetic N-iPSC?
- 3d: The use of terminology is inconsistent here. Please label these cells as N-CB-ipSC rather than +CB-iPSC, and N-D-iPSC rather than +D-iPSC to avoid confusion. This should also be consistent in other figures and throughout the manuscript.

Figure 4

- 4a, c, d, e: Please show statistics where appropriate
- 4a: What are RUES01 and E1CA2? This is not explained anywhere in the manuscript.
- 4c,d: Please explain the significance of this assay.
- 4f: In the manuscript text, please specify that H9, C1.2 and C2 are non-diabetic hiPSC-VP, while E1CA1, E1CA2 and C1C1 are diabetic hiPSC.

Figure 5

- WBs assessment should be quantified.

Figure 6

- 6d, e: Please show statistics, and label the top of the graphs with the time points
- 6e: This panels is not cited in the main text

Figure 7

- 7a-c: These panels are not cited in the main text
- 7d,f: Please label these images "LIF-3i-VP" such that it is clear what the images represent
- This experiment should also be conducted using non-diabetic hiPSCs

Figure 8

- 8d-f: Please show statistics

General suggestions

- The authors jump to using many different stem cell types without justifying the rationale or providing any explanation of the model. For example, figure 3 uses the APEL system and cord blood-derived hiPSC lines, and the authors do not explain why these various models are selected, or what their significance is. Although word limitations may be problematic, please introduce and justify that rationale of each model whenever possible to avoid confusion and help readers understand the significance of the findings.
- In many cases, for example Fig. 3d, the authors compare D-iPSCs from fibroblasts with iPSCs from other sources, but do not make direct comparisons to non-diabetic fibroblast-derived iPSCs. What is the rationale for this?
 - o Comparisons to non-diabetic fibroblast iPSCs should be included.
- In figure legends, please specify the "n" and statistics used for each experiment
- In many data figures, for example fig. 7, no statistics are shown. Please either specify the rationale for not conducting statistical analyses or show statistics.
- Please specify the statistical tests used in the figure legends.

Compositional suggestions

- There are many abbreviations, which make the manuscript difficult to follow. Please use abbreviations conservatively. A list of abbreviations, if allowed by journal standards, would also be helpful.
- Please keep the terminology consistent throughout the manuscript. For example, use either "Naïve" or "LIF-3i." This will help to avoid confusion and reduce the number of abbreviations used.
- Please better introduce the ischemic retinopathy model in the main text. Most readers will not be familiar with the model, as this is a general journal.

Reviewer #2 (Remarks to the Author):

In this manuscript by Park et al, the authors compared the differentiation between an alternative tankyrase inhibitor-regulated human naïve PSC (LIF-3i N-hiPSC) and conventional primed PSC. Teratoma assay revealed that LIF-3i N-hiPSC differentiated into three germ layers more efficiently than isogenic primed hiPSC. Next, authors focused on the differentiation into vascular progenitors (VP) and VP function using diabetic patient's iPSC. LIF-3i PSC generated more VP cells than primed PSC. Additionally, VP cells from LIF-3i had better proliferation, lower senescence, less DNA damage, and more efficient engraftments than VP from primed. Authors also explained hypomethylation and less expression of lineage-specific genes in LIF-3i caused this advantage of VP cells.

Authors' claims are reasonable and important for the field in regenerative medicine using iPSC. The study would be strengthened by consideration of the following issues.

Major points

1. Authors mainly compared LIF-3i iPSC and isogenic primed iPSC. I was wondering whether their claim could be extended to ESC. Do LIF-3i ESC have functional advantages, especially in figure 1 (teratoma) and 6 (transplantation)?
2. Authors explained that functional advantages of LIF-3i VP were the outcomes of the gene expression and epigenetic state at the stage of PSC. Although authors checked a couple of surface markers and genes of VP population, I believe it is necessary to compare the global gene expression of VP derived from LIF-3i and primed.

Minor points

1. Figure S1b: There are no labels of green, red, round etc. In the main text, authors mentioned

FOXA2+. But there is no FOXA2 image.

2. Immunostaining images of Figure 2a: There are no controls. It will be better to show Figure 2a both by immunostaining and by western blotting.

3. FigS3c: Do VP express SSEA4?

4. Sentence of main text p8 L9-14 (Figs4, S4a...) needs to be corrected.

Reviewer #3 (Remarks to the Author):

This timely and interesting study by Park et. al. shows that the obstacle of incomplete reprogramming and lineage priming of conventional fibroblast-hiPSC can be overcome with molecular reversion to a tankyrase inhibitor-regulated naïve epiblast-like state with a more primitive, unbiased epigenetic configuration. Vascular progenitors (VP) differentiated from naïve patient-specific hiPSC maintained greater genomic stability than VP generated from conventional hiPSC, and had higher expression of vascular identity markers, decreased non-lineage gene expression, and were superior in migrating to and re-vascularizing the deep neural layers of the ischemic retina. The authors show data that conventional skin fibroblast-derived hiPSC lines had higher rates of reprogramming errors and displayed poorer vascular differentiation and in vivo retinal engraftment efficiencies relative to conventional CB-hiPSC and that their novel process of LIF-3i modification removed these differences between the cell of origin of the hiPSCs. The studies using diabetic donors and the epigenetic studies are very novel and highly important.

The beginning of the manuscript is a bit difficult to read and understanding of the generation of the N-hiPSCs is not explained well enough that an interested reader outside the iPSC field could understand how LIF and three small molecules inhibiting the tankyrase, MEK, and GSK3b signaling pathways (LIF-3i) had the effect they did (without going to earlier references and reading them in detail).

The significance of inhibition of tankyrase-PARP activity is never really explained for the reader.

The significance of naïve-specific proteins (KLF2, NR5A2, TFCP2L1, STELLA/DPPA3, E-CADHERIN) is not explained. The authors are very familiar with the literature. At times they do not provide the reader with enough information (in the actual manuscript) to appreciate the importance of the work presented.

The authors state "LIF-3i reversion permitted comparable efficiencies of generation of CD31+CD146+CXCR4+ VP populations regardless of conventional hiPSC donor source." However, it is unclear how many fibroblast-derived vs. cord blood derived N-hiPSCs were actually examined to make this bold statement. Please clarify.

Figure 2: The nomenclature for the lines is confusing – is E5C3 a diabetic line?

Figure 4: Stats for 4a, c, and e are missing.

Figure 5: Stats are not provided for Figure 5 C.

Figure 6c- the letter "c" is missing from the legend but in the figure the HNA + cells that are shown to be "engrafted" should be co labeled with either an endothelial or pericyte specific marker as it is difficult to tell what they have become if anything. Higher magnification to actually show incorporation are needed. Stats are missing for 6d and e.

Figure 7: How much of the blood vessel had to actually be CD34+ before it was considered in the analysis for Figure 7b?

Maria Grant

October 20th, 2019

RESPONSE TO REVIEWERS

REVISED RE-SUBMISSION (NCOMMS-19-18543)

“Vascular Progenitors Generated from Tankyrase Inhibitor-Regulated Naïve Diabetic Human iPSC Potentiate Efficient Revascularization of Ischemic Retina” (revised title).

In this work, we report the first translational application of a new class of tankyrase inhibitor-regulated naïve human induced pluripotent stem cells (N-hiPSC) for improving vascular regenerative therapies in diabetes ¹. To date, there has not been any human naïve pluripotent stem cell system that has been demonstrated to be effective for pre-clinical cellular therapies, and to our knowledge this study is the first one.

Several groups have reported various complex small molecule approaches that putatively captured human ‘naïve-like’ pluripotent molecular states that are more primitive than those exhibited by conventional, lineage-primed hiPSC ². However, many culture systems supporting human naïve-like pluripotency have potentiated karyotypic instability, global loss of parental genomic imprints, and impaired multi-lineage differentiation performance ². We first reported in 2016 that tankyrase-inhibited N-hiPSC and N-hESC did not suffer these caveats of other human naïve reversion methods ³. These naïve human pluripotent stem cells (N-hPSC) possessed greater differentiation potency than conventional hPSC ³; this improved functional pluripotency was potentiated by inclusion of the small molecule tankyrase/PARP (poly ADP ribose polymerase) inhibitor XAV939 to the classical 2i (GSK3b/MEK inhibition) naïve reversion chemical cocktail (LIF-3i). Our report herein, is the first to confirm that, along with expanded pluripotency stem cells (EPSC) that also employ XAV939 chemical modulation ^{4,5}, tankyrase-inhibited naïve hiPSC are members of a *bona fide* new class of pluripotent stem cells with high functionality and improved epigenetic stability that will highly impact regenerative medicine.

The mechanism of tankyrase (XAV939)-inhibited augmentation of ‘expanded’ pluripotency ^{4,5} is now under intense investigation by many investigators, and this work should have broad interest to the wide audience of *Nature Communications*. For example, herein, we revealed a putative epigenetic mechanism of improved tankyrase inhibitor-regulated N-hiPSC functionality. We demonstrated that CpG DNA methylation and histone configurations at developmental promoters of diabetic tankyrase inhibitor-regulated N-hiPSC possessed tight regulation of lineage-specific gene expression and a de-repressed naïve epiblast-like epigenetic state that was highly poised for multi-lineage transcriptional activation. We propose that autologous or cell-banked vascular/pericytic progenitors derived from tankyrase inhibitor-regulated N-hiPSC will more effectively reverse the epigenetic pathology that drive disorders such as diabetes. We are confident that the application of this new class of human stem cells will inspire new directions of investigation for understanding human pluripotency, and for improving the utility of hiPSC therapies in regenerative medicine.

In response to the thoughtful and detailed critiques of our Reviewers, we have performed new experiments, and *extensively* revised the manuscript to provide all the edits and clarifications requested by our Reviewers. Additionally, we have revised the title, focused the writing, and clarified the presentation of the figures to sharpen the message of our findings. These major revisions have improved and now make the work more accessible to the general audience of *Nature Communications*. Below is a detailed point-by-point address of the comments and critiques provided to us by our three Reviewers. Specific revisions and clarifications in response to our Reviewers are cited, and then identified in the revised text, using tracked **green** font.

Reviewer 1 Comments:

The authors present a novel and highly relevant manuscript, in which they utilize naïve human induced pluripotent stem cells (N-iPSCs) from diabetic donors to facilitate recovery of vascular injury in an ischemic model of retinopathy. This is relevant to diabetic retinopathy, as vascular damage is a primary disease process. The use of reprogrammed fibroblasts to repair vascular damage has therapeutic potential for diabetic retinopathy, and potentially for other diabetic complications and retinopathies of other etiologies. The authors convincingly demonstrate that N-iPSCs have a less-differentiated phenotype than conventionally prepared iPSCs. Their findings also suggest that iPSCs reprogrammed with this method have a more reparative phenotype, even when the cells are derived from diabetic donors, as would be the case in the context of diabetic retinopathy. The manuscript has a few weaknesses that should be addressed. In its current form, the manuscript is very difficult to follow due to the extensive use of abbreviations, and inconsistent use of terminologies. Please re-read the manuscript carefully and ensure that all terminologies are consistent. Some of the models used are not described or justified, as specified below. Finally, some of the results appear to lack requisite statistical analyses and essential controls, as specified in the figure critique and recommended additional experiments.

We are grateful to Reviewer 1 for constructive and remarkably detailed critiques, and take the opportunity to clarify our claims. We have extensively re-written our manuscript using specific comments made below. All relevant figures in this paper have now been majorly revised to improve clarity, as recommended. Specific critiques are addressed below.

Figure 1. The use of “E8” is confusing, as this is not explained in the results section. Please either explain that this signifies E8 media, or label the figure differently, for example with “control.”

Thank you for this suggestion. All figures in this paper (where relevant) have now been re-labeled to improve clarity, as recommended. For example, hPSC cultures grown in conventional, primed E8 media culture conditions are now re-labeled as “Primed” or “P”, and all hiPSC cultures grown in naïve-reverted LIF-3i conditions are now re-labeled as “Naïve” or “N”. Additionally, diabetic vascular progenitors are labeled as DVP, and naïve diabetic vascular progenitors are labeled N-DVP. We have extensively revised the entire manuscript, all the figures, and all the figure legends to insure a consistent use of these abbreviations to increase the clarity of presentation.

Figure 2. Where possible a quantitative assessment of images should be included in the main figures.

As requested, a new supplementary datasheet (**Table S4**) was created with densitometry quantitation tables of all Western blots shown in this manuscript (normalized to their actin controls using ImageJ software). Please also note that as required for publication in *Nature* journals, we have also included the images of all raw, uncropped Western blots in this paper in a zip-compressed ‘Source Data’ supplementary file, that includes the raw data and statistical worksheets. Thank you.

Figure 3b-d: Please show statistics. 3b-c: Please include legends for graphs.

Done where relevant; please see revised Fig3b,c and Fig3b,c figure legends that include revised statistical test information. Thanks.

Fig. 3d (revised Fig 3c): Please explain the rationale for comparing to CB-iPSC. What is the efficiency in diabetic versus non-diabetic N-iPSC?

Thank you for the opportunity to clarify this critical point in detail. In brief, we have previously established ^{2,3,6} that the most appropriate hiPSC control for a LIF-3i naïve reversion experiment is not another, independent, ‘similar’ hiPSC line, but the same, genotypic-identical (**isogenic**) primed hiPSC line that was used for reversion to N-hiPSC. An ample body of literature has demonstrated that there is great and unpredictable variability in the differentiation potency from one primed conventional hiPSC line to another due to multiple, undefined variables ⁷⁻¹⁸. In various experiments in this work, we compared functional results of independent primed and naïve DhiPSC to non-diabetic fibroblast-derived N-hiPSC and non-diabetic N-CB-hiPSC, and demonstrated that the efficiency of VP generation is not significantly different. However, the most accurate control in each case for a naïve-reverted N-hiPSC line is *its isogenic, genotypic-identical, primed DhiPSC counterpart from which it was directly derived*. Every experiment in this work includes, at a minimum, comparisons of isogenic primed vs naïve DhiPSC, in addition to comparisons to various ‘normal, non-diabetic primed vs naïve hiPSC (derived from independent donors).

To better sharpen the presentation of this experimental design, we have revised the Methods section to emphasize better the isogenic hiPSC experimental design (pp 20-210). We have ALSO added a new schematic in Fig. 1a to clarify the design for general readers not familiar with the literature. We also discuss comparative implications of naïve reversion in the context of VP derived from both normal CB-derived N-hiPSC, normal fibroblast-derived N-hiPSC, and diabetic N-DhiPSC on pp. 14-15 of the Discussion section of the paper.

As further background, this isogenic experimental design derives from extensive previous work that we ^{1,2,3,6} and others ⁷⁻¹⁸ have established demonstrating that conventional, primed hiPSC (and also hESC) display wide and often dramatic interline multi-lineage differentiation variability, especially from one primed fibroblast-hiPSC line to another ⁷⁻¹². Although reprogramming-associated errors ¹³⁻¹⁵ and retention of donor cell-specific epigenetic memory ^{16,17} may be responsible for some of this differentiation bias from one primed conventional hiPSC to another, donor-specific genetic variability affecting lineage-primed gene expression may play the more dominant role ¹⁸. However, regardless of mechanism, our previous work (Zimmerlin et al, 2016) revealed that conventional hiPSC interline variability of differentiation and lineage-primed gene expression (from either primed fibroblast-hiPSC or primed CB-hiPSC) was effectively diminished following reversion to a naïve pluripotent state with the LIF-3i system ³. We also recently published a review on this important topic to put these results in perspective to the literature. ²

Furthermore, the finding of superiority of efficiency of VP generated from primed CB-iPSC was first reported in our original study published in *Circulation* (Park et al, 2014) ¹, which this study is an extension of. In that paper, we reported that VP derived from high-quality conventional, primed CB-iPSC possessed significantly augmented capacity for regenerating ischemic retinal vasculature, compared to those from conventional fibroblast-hiPSC. In that paper, we also established that this was due to the phenomenon that CB-iPSC possessed relatively less lineage priming, less “epigenetic memory”, and higher vascular differentiation potency than most fibroblast-derived hiPSC. Reversion of a conventional hiPSC to a naïve pluripotent state takes this principle even farther ³.

Thus, in these (as well as our previous studies with normal (non-diabetic) hiPSC^{12,13}), to examine the differentiation competence of both normal non-diabetic and diabetic N-hiPSC, and to minimize hiPSC variations within directed differentiation experiments that arise from interline variability and genetic aberrations of non-genotypic-identical primed hiPSC differences not related to diabetic epigenetic lesions, we directly differentiated naïve vs their primed sibling isogenic DhiPSC counterparts in a parallel manner. Paired isogenic primed and LIF-3i-reverted normal hiPSC and DhiPSC lines of identical passage were simultaneously and directly cultured into defined, feeder-free differentiation systems according to manufacturer's directions.

Thus, we employ not only an isogenic experimental design, but also utilize both normal CB-iPSC-derived and normal fibroblast hiPSC-derived VP as "additional controls" to isogenic diabetic donor DhiPSC-derived DVP vs N-DVP (e.g., in Fig. 3 and other figures). For example, we include not only the results of isogenic primed vs naïve DhiPSC, but also the performance of known "high-performing" CB-iPSC lines (e.g., E5C3) or "high-performing fibroblast" hiPSC lines (e.g., C1.2, C2). These normal non-diabetic lines have excellent track records for robust VP generation from our previous publications in *Circulation*¹ and *Development*³, and serve as excellent controls for disease-derived DhiPSC that may not demonstrate normal VP generation, even after naïve reversion.

Fig. 3d: The use of terminology is inconsistent here. Please label these cells as N-CB-iPSC rather than +CB-iPSC, and N-D-iPSC rather than +DiPSC to avoid confusion. This should also be consistent in other figures and throughout the manuscript.

Done, as advised. The previous Figure 3d (now Fig 3c) and other similar sections have been revised, as recommended, for designating naïve vs primed CB-iPSC and DhiPSC.

Fig. 4a, c, d, e: Please show statistics where appropriate.

Figure 4 has been completely revised. Statistical information has now been included in detail in the figure legend, and data has been re-graphed where relevant to show independent measurements of each assay ("n").

Fig. 4a: What are RUES01 and E1CA2? This is not explained anywhere in the manuscript.

The text of the manuscript and the figure legends have all now been revised to indicate the identity of hESC lines (e.g., RUES01, H9) and hiPSC lines (e.g. diabetic line E1CA2), wherever used in an experiment. Furthermore, the sources and identities of all hPSC lines used in this paper are all detailed and catalogued in a revised **Table S1**, which is cited amply throughout the text. Thanks.

Fig. 4c,d: Please explain the significance of this assay.

Figure 4 has been extensively revised. The significance of the endothelial Dil-acetylated-LDL uptake assay, is that it is a common assay used to demonstrate endothelial cell functionality. We have revised the Methods section (p. 24, "vascular function assays") and the Results section (p. 9) to make the purpose and use of this assay clearer. Thank you.

Fig. 4f: In the manuscript text, please specify that H9, C1.2 and C2 are non-diabetic hiPSC-VP, while E1CA1, E1CA2 and C1C1 are diabetic hiPSC.

As explained above, the text of the manuscript and the figure legends have all now been revised to indicate the identity of hESC lines (e.g., RUES01, H9) and hiPSC lines (e.g., diabetic line E1CA2) wherever used in an experiment. The sources and identities of all hPSC lines used in this paper are all detailed in revised **Table S1**. Thank you.

Figure 5; WBs assessment should be quantified.

Same answer as above. A new supplementary datasheet (Table S4) was created with densitometry quantitation tables of all Western blots shown in this manuscript (normalized to their actin controls using ImageJ software). Thank you.

Figure 6d, e: Please show statistics, and label the top of the graphs with the time points. 6e: This panel is not cited in the main text

Figure 6 has been completely revised. Labeling of all figures and panels was done as recommended. All statistics are now shown in both the figures and figure legends. Thanks.

Figure 7a-c: These panels are not cited in the main text.

Corrected, thanks.

Fig. 7d,f: Please label these images “LIF-3i-VP” such that it is clear what the images represent. This experiment should also be conducted using non-diabetic hiPSCs. Figure 8d-f: Please show statistics.

The terminology has now been simplified for reader clarity. The labeling now used all throughout the revised manuscript is now ‘primed DVP’ (for DVP derived from primed diabetic hiPSC) and ‘N-DVP’ (for DVP derived from naïve DhiPSC). The accuracy of using isogenic hiPSC (and not another, genetically-independent, non-diabetic hiPSC) as the most accurate control of primed vs naïve hiPSC experiments was described in detail above. As described above, based on our and others’ published results, the use of genetically-independent primed hiPSC as controls for normal or diseased hiPSC lines, can lead to misleading interpretation due to the highly variable multi-lineage differentiation efficiency of one primed hiPSC line to another. This is especially true for VP differentiation from primed fibroblast-hiPSC > primed CB-hiPSC lines, as we previously published¹. SEM’s for all PCR experiments in Fig. 8 are shown. Thanks

General suggestions

• **The authors jump to using many different stem cell types without justifying the rationale or providing any explanation of the model. For example, figure 3 uses the APEL system and cord blood-derived hiPSC lines, and the authors do not explain why these various models are selected, or what their significance is. Although word limitations may be problematic, please introduce and justify that rationale of each model whenever possible to avoid confusion and help readers understand the significance of the findings. In many cases, for example Fig. 3d, the authors compare D-iPSCs from fibroblasts with iPSCs from other sources, but do not make direct comparisons to non-diabetic fibroblast-derived iPSCs. What is the rationale for this? Comparisons to non-diabetic fibroblast iPSCs should be included.**

Thank you, as discussed above, we have now extensively revised the paper as recommended to address these issues. The rationale for the experimental design utilized in these studies is summarized in detail above.

• **In figure legends, please specify the “n” and statistics used for each experiment.**

Corrected, thanks. We have re-graphed most figures all throughout to show scatter dot-bar plots and scatter dot-box-whisker plots of individual measurements, so that “n” is directly shown for. Thanks.

- In many data figures, for example fig. 7, no statistics are shown. Please either specify the rationale for not conducting statistical analyses or show statistics.

Corrected throughout all figures. Statistics has been included where relevant.

- *Please specify the statistical tests used in the figure legends.*

Corrected throughout. Statistical info has now been included where relevant in all figure legends.

Compositional suggestions: There are many abbreviations, which make the manuscript difficult to follow. Please use abbreviations conservatively. A list of abbreviations, if allowed by journal standards, would also be helpful.

Thanks for this suggestion, we have revised the paper to minimize abbreviations. We will also inquire with the Editor about the possibility of including a list of abbreviations.

Please keep the terminology consistent throughout the manuscript. For example, use either “Naïve” or “LIF-3i.” This will help to avoid confusion and reduce the number of abbreviations used.

Thank you. As already outlined above, to increase the clarity of the paper’s presentation, we majorly revised the manuscript. Primed E8 media culture conditions are now re-labeled as “Primed” or “P”, and all hiPSC cultures grown in naïve-reverted LIF-3i conditions are now re-labeled as “Naïve” or “N”.

Please better introduce the ischemic retinopathy model in the main text. Most readers will not be familiar with the model, as this is a general journal.

Thank you for this great suggestion. We have expanded the text of Results section on pp. 9-10 where we introduce the model, and provided more background, explanation, and references of the rationale for use of this I/R model that we employed to test the functionality of naïve vs primed DVP. We have also revised Figure 6 to include a new schematic that explains the I/R model better, and how we use it in this paper (*i.e.*, Fig. 6a).

Reviewer 2 Comments:

In this manuscript by Park et al, the authors compared the differentiation between an alternative tankyrase inhibitor-regulated human naïve PSC (LIF-3i N-hiPSC) and conventional primed PSC. Teratoma assay revealed that LIF-3i N-hiPSC differentiated into three germ layers more efficiently than isogenic primed hiPSC. Next, authors focused on the differentiation into vascular progenitors (VP) and VP function using diabetic patient’s iPSC. LIF-3i PSC generated more VP cells than primed PSC. Additionally, VP cells from LIF-3i had better proliferation, lower senescence, less DNA damage, and more efficient engraftments than VP from primed. Authors also explained hypomethylation and less expression of lineage-specific genes in LIF-3i caused this advantage of VP cells. Authors’ claims are reasonable and important for the field in regenerative medicine using iPSC. The study would be strengthened by consideration of the following issues.

We are grateful for Reviewer 2's constructive critiques. We have majorly revised our data and presentation according to this Reviewer's suggestions to improve areas of apparent lack of clarity, as outlined below. We have also conducted major new bioinformatics experiments to specifically respond to Reviewer 2's queries.

Major points

1. Authors mainly compared LIF-3i iPSC and isogenic primed iPSC. I was wondering whether their claim could be extended to ESC. Do LIF-3i ESC have functional advantages, especially in figure 1 (teratoma) and 6 (transplantation)?

Yes, we recently published ^{3,6} that the LIF-3i method does augment multi-lineage (mesoderm, ectoderm, endoderm) functional pluripotency, and does confer functional differentiation advantages for a panoply of both conventional primed hESC (e.g. for hESC line H9 ³) and conventional, primed hiPSC. In our original work (Zimmerlin et al, *Development*, 2016) ³, and in our methods paper describing the detailed protocol for LIF-3i naïve reversion of hESC and hiPSC (Park et al *JoVE*, 2018) ⁶, we demonstrated its efficacy in a broad repertoire of over 20 hESC and hiPSC, each derived from genotypic-independent donors. This paper extends the LIF-3i naïve reversion system described in these preceding papers, and focuses primarily on improving functional pluripotency in disease-derived diabetic hiPSC. In any case, we do use multiple N-hiPSC reprogrammed from fibroblasts and cord blood cells as well as include N-hESC in this work (e.g., H9, RUES02- see Fig. S1d) to demonstrate the efficacy of vascular differentiation improvement in not only normal non-diabetic hiPSC and normal hESC. Additionally, in response to this query, we conducted extensive new bioinformatics RNA-Seq experiments that include primed VP vs. N-VP from not only DhiPSC lines but also normal non-diabetic hiPSC and normal hESC (see below).

2. Authors explained that functional advantages of LIF-3i VP were the outcomes of the gene expression and epigenetic state at the stage of PSC. Although authors checked a couple of surface markers and genes of VP population, I believe it is necessary to compare the global gene expression of VP derived from LIF-3i and primed.

Thank you for this important recommendation. As advised and directed by Reviewer 2, we have performed and now include new RNA-Seq experiments to define the global differential gene expression profiles of primed VP vs. N-VP from both DhiPSC and normal non-diabetic hESC and hiPSC (**Fig. S8**).

Minor points

1. Figure S1b: There are no labels of green, red, round etc. In the main text, authors mentioned FOXA2+. But there is no FOXA2 image.

Thank you. As recommended, the graphs in Figure S1 have been extensively revised, re-organized, and the figure legends extensively re-written to improve the clarity of presentation. Endodermal differentiations are now shown in Fig. S1c. Shown are definitive endodermal FOXA2⁺ differentiations of independent isogenic primed vs naïve-hiPSC ($n=6$; 3 normal fibroblast-hiPSC and 3 normal CB-hiPSC lines, described in Table S1). The differentiation result of each individual primed (blue) or naïve (red) isogenic hiPSC line is represented by a different shape (round, triangle, etc), with means and SEM's of all lines shown. Differentiations were performed with STEMdiff definitive endoderm kit and STEMdiff APEL medium (StemCell Technologies), as previously described ¹².

2. Immunostaining images of Figure 2a: There are no controls. It will be better to show Figure 2a both by immunostaining and by western blotting.

Immuno-stains of N-DhiPSC for nuclear pluripotency transcription factors (e.g., NANOG and KLF2) are more accurate than Western blots in their interpretation, when demonstrated to be *nuclear* (i.e., protein expression is shown to co-localize with DAPI). Clear demonstration of nuclear expression in the native cellular morphology of transcription factors is best performed with an IF experiment, since Western blot analysis of protein lysates can produce false positive bands of antibody cross-reacting cytoplasmic protein. The IF controls for these IP experiments are secondary antibodies or isotype control antibodies which were all negative, not shown, and performed routinely in every experiment. Additionally, the nuclear factor over-expressions of naïve factors in LIF-3i-reverted in N-DhiPSC were previously validated with IF in our previous publications of normal non-diabetic hESC/hiPSC, and shown to correlate directly with gene array and qPCR over-expressions^{3,6}. Similarly, expression of the adhesion molecule E-Cadherin is most convincing when clearly demonstrated in the extracellular space, as shown in Fig. 2a. However, we do *also* show Western blots of selected naïve cytoplasmic and nuclear factors such as **phosphorylated-STAT3** (Fig. 2b) and **TFAP2C**¹⁹ (Fig. 2c), which are two of the most specific characteristics of a naïve pluripotent state in both murine ESC and hPSC.

3. FigS3c: Do VP express SSEA4?

Yes, both primed and naïve CD31⁺CD146⁺ VP have an embryonic (not adult) phenotype, and therefore express SSEA4. We show this in Figure S3c. We previously reported this novel hPSC-derived population as highly proliferic pericytic-vascular stem-progenitors¹. As is shown, both primed VP and N-VP also express abundant amounts of a myriad of other embryonic/vascular stem progenitor markers (e.g., CD34, CXCR4, CD90, CD105, and CD144).

4. Sentence of main text p8 L9-14 (Figs4, S4a...) needs to be corrected.

Thank you, corrections made.

Reviewer 3 Comments:

This timely and interesting study by Park et. al. shows that the obstacle of incomplete reprogramming and lineage priming of conventional fibroblast-hiPSC can be overcome with molecular reversion to a tankyrase inhibitor-regulated naïve epiblast-like state with a more primitive, unbiased epigenetic configuration. Vascular progenitors (VP) differentiated from naïve patient-specific hiPSC maintained greater genomic stability than VP generated from conventional hiPSC, and had higher expression of vascular identity markers, decreased non-lineage gene expression, and were superior in migrating to and re-vascularizing the deep neural layers of the ischemic retina. The authors show data that conventional skin fibroblast-derived hiPSC lines had higher rates of reprogramming errors and displayed poorer vascular differentiation and in vivo retinal engraftment efficiencies relative to conventional CB-hiPSC and that their novel process of LIF-3i modification removed these differences between the cell of origin of the hiPSCs. The studies using diabetic donors and the epigenetic studies are very novel and highly important.

We are grateful to Reviewer 3 for constructive critiques, and take the opportunity to clarify our claims. We have extensively re-written our manuscript using specific comments made by our Reviewer. We have responded with major revisions of our data presentation and re-written selected sections to improve clarity.

The beginning of the manuscript is a bit difficult to read and understanding of the generation of the N-hiPSCs is not explained well enough that an interested reader outside the iPSC field could understand how LIF and three small molecules inhibiting the tankyrase, MEK, and GSK3b signaling pathways (LIF-3i) had the effect they did (without going to earlier references and reading them in detail). The significance of inhibition of tankyrase-PARP activity is never really explained for the reader. The significance of naïve-specific proteins (KLF2, NR5A2, TFCP2L1, STELLA/DPPA3, E-CADHERIN) is not explained. The authors are very familiar with the literature. At times they do not provide the reader with enough information (in the actual manuscript) to appreciate the importance of the work presented.

We have revised and beefed up the abstract and the Introduction (revisions in green font) that describe the chemical naïve reversion of hPSC with more information of the salient conclusions of our previously published and well cited novel LIF-3i naïve reversion system^{3,6}, to provide a better background to the general reader. The significance of the tankyrase-PARP activity in stabilizing the human naïve state and augmenting functional pluripotency was the main topic of our previously published *Development* paper³. We also cite an extensive review we recently published² that puts the results of our LIF-3i system in perspective to the literature. We reference this review amply throughout the paper. We have also majorly revised the Results section on p.9, 1st paragraph to clarify the importance of the naïve-specific protein factors in defining a *bona fide* naïve epiblast-like state. We have also revised our Discussion on p. 16 to clarify the proposed epigenetic mechanism by which the tankyrase/PARP inhibitor XAV939 mediates augmented functional pluripotency and superior VP generation. Finally, we have constructed and included a new schematic (**Fig. 9**) that summarizes the main epigenetic mechanistic findings in this paper as it relates to LIF-3i naïve reversion. Thanks for this critique which prompted us to revise and clarify the paper.

The authors state “LIF-3i reversion permitted comparable efficiencies of generation of CD31+CD146+CXCR4+ VP populations regardless of conventional hiPSC donor source.” However, it is unclear how many fibroblast-derived vs. cord blood derived N-hiPSCs were actually examined to make this bold statement. Please clarify.

We have toned down this claim, and clarified what we actually meant to convey for the results presented in Fig. 3c on p.8: “... naïve-reverted fibroblast-derived N-DhiPSC and non-diabetic cord blood (CB)-derived N-CB-hiPSC lines both generated similar efficiencies of VP, despite a previously reported higher VP differentiation efficiency of conventional CB-hiPSC compared to conventional fibroblast-derived hiPSC...” Thanks for this constructive critique.

Figure 2: The nomenclature for the lines is confusing – is E5C3 a diabetic line?

No, E5C3 is a CB-iPSC line with high-performing capacity to generate VP¹. We previously published E5C3 and other similar CB-iPSC lines and their multi-lineage differentiation performance before and after naïve reversion^{2,3}. The text of the manuscript, all the figures (including Fig 2), and the figure legends have now been revised to make clear the identity of all hPSC lines used in these studies, including E5C3. For example, we have clarified (as per previous questions above) the identity of hESC lines (e.g., RUES01, H9), and the normal and diabetic hiPSC lines wherever used in an experiment in the figure legends. For clarity, all figures were re-labeled not with the name of the individual cell line, but with generic labels such as ‘hiPSC’ (i.e., primed normal hiPSC), N-hiPSC (i.e., naïve hiPSC), DhiPSC (i.e., diabetic primed hiPSC), or N-DhiPSC (i.e., naïve diabetic hiPSC). The cell line name is cited in the figure legend. Most importantly, the sources and identities of all hPSC lines used in this paper are all detailed and catalogued in a revised **Table S1**, which is cited amply throughout the text. Thanks.

Figure 4: Stats for 4a, c, and e are missing.

Figure 4 has been majorly revised and all relevant statistics are now included in the figure and figure legend. Thank you.

Figure 5: Stats are not provided for Figure 5C.

Relevant statistics are now included in the figure and figure legend. Thank you.

Figure 6c- the letter “c” is missing from the legend but in the figure the HNA + cells that are show to be “engrafted” should be co labeled with either an endothelial or pericyte specific marker as it is difficult to tell what they have become if anything. Higher magnification to actually show incorporation are needed. Stats are missing for 6d and e.

Thanks. Data for Figure 6, its legend, and relevant statistics are now all majorly revised to improve clarity. We provide both low and high magnification images of engrafted chimeric human-murine vessels in the neural retina in a revised Figure 7. Importantly, we wish to note that both primed VP and N-VP both already possess a pericytic-endothelial phenotype *a priori* following their generation from our hPSC modified APEL vascular differentiation protocol, *i.e.*, prior to their intra-vitreous injections *in vivo* (Fig. 3a, lower panel). For example, we show in Fig. S3c that both primed DVP and N-DVP possess embryonic pericytic-endothelial phenotypes (*i.e.*, CD31⁺CD146⁺CXCR4⁺CD44⁺CD144⁺SSEA4⁺CD45⁻ surface expressions). However, we do agree that it is a very important question whether engrafted N-VP maintain pericyte phenotypes *after* engraftment (and potentially even undergo pericyte stem cell self-renewal *in vivo* following engraftment in retinal vessels) However, that aim is beyond the scope of the current manuscript. As an aside, unlike our use of highly reliable and specific antibodies for human CD34 and human CD31 in these studies, determining human pericyte-specific phenotype maintenance/self-renewal following engraftment has not been technically easy for us, since there are not good antibodies for human-specific pericyte markers (*e.g.* human-specific NG2 antibodies) that are capable of distinguishing human vs murine pericytes *in vivo*. Co-staining with HNA has not provided adequate cytoplasmic resolution of engrafted cells since although very human-specific, anti-HNA antibody labels a nuclear histone protein, and shows only presence of human nuclei. Future experimental designs will aim to resolve this goal by utilizing N-VP with fluorescent genetic tagging (*e.g.*, with tdTomato protein expression in cytoplasm) along with co-localization using anti-murine/human pericyte antibodies, to distinguish the presence of human pericytes within murine vessels, following long-term chimeric engraftment.

Figure 7: How much of the blood vessel had to actually be CD34+ before it was considered in the analysis for Figure 7b?

Thanks. Briefly, human-specific CD34 expression (**Fig S7b**) or human-specific CD31 expression was scored when the Alexa488⁺-conjugated signal was expressed at chimeric human-murine blood vessels that also expressed murine collagen IV (mColIV) or murine CD31 (mCD31) (**Fig, 7a,c**). The numbers of human blood vessels detected in murine vessels ranged between 1-13 per image at 20X objectives (**Fig, 7b,d**). The Methods section (p.30) and **Fig S7** have been revised to provide more details on these quantification approaches.

We once again thank our Reviewers for their thoughtful critiques for improving our manuscript, and we are confident that we have now clarified all queries. We are hopeful that this manuscript is now ready for publication in *Nature Communications*.

References

1. Park, T.S. et al. Vascular progenitors from cord blood-derived induced pluripotent stem cells possess augmented capacity for regenerating ischemic retinal vasculature. *Circulation* 129, 359-372,
2. Zimmerlin, L., Park, T. S. & Zambidis, E. T. Capturing Human Naive Pluripotency in the Embryo and in the Dish. *Stem Cells Dev* 26, 1141-1161, doi:10.1089/scd.2017.0055 (2017).
3. Zimmerlin L, Park TS, Huo JS, Verma Talbot Jr. CC, Agarwal J, Steppan D, Peters A, Zang Y, Guo H, Pandiyan K, Zhong X, Guiterez G, Hampton C, Young C, Canto-Soler V, Friedman A, Baylin SB, Zambidis ET, Tankyrase inhibition promotes a stable human naïve pluripotent state with improved functionality. *Development* 2016; 143(22): 4368-4380.
4. Yang, J., et al., Establishment of mouse expanded potential stem cells. *Nature*, 2017. **550**(7676): p. 393-397.
5. Gao et al, Establishment of porcine and human expanded potential stem cells. *Nat. Cell Biol.* (2019). 21, 687-699.
6. Park, T. S., Zimmerlin, L., Evans-Moses, R. & Zambidis, E. T. Chemical Reversion of Conventional Human Pluripotent Stem Cells to a Naive-like State with Improved Multilineage Differentiation Potency. *J Vis Exp*, doi:10.3791/57921 (2018).
7. Osafune K, et al. (2008) Marked differences in differentiation propensity among human embryonic stem cell lines. *Nature Biotechnol* 26, 313-315.
8. Choi KD, et al. (2009) Hematopoietic and endothelial differentiation of human induced pluripotent stem cells. *Stem Cells* 27, 559-567.
9. Feng Q, et al. (2009) Hemangioblastic derivatives from human induced pluripotent stem cells exhibit limited expansion and early senescence. *Stem Cells* 28, 704-712.
10. Hu, B. Y. et al. (2010) Neural differentiation of human induced pluripotent stem cells follows developmental principles but with variable potency. *Proc Natl Acad Sci USA* 107, 4335-4340.
11. Bock C, et al. (2011) Reference Maps of human ES and iPS cell variation enable high-throughput characterization of pluripotent cell lines. *Cell* 144, 439-452.
12. Boulting GL, et al. (2011) A functionally characterized test set of human induced pluripotent stem cells. *Nature Biotechnol* 29, 279-286.
13. Nishino, K. et al. (2011) DNA methylation dynamics in human induced pluripotent stem cells over time. *PLoS Genet* 7, e1002085.
14. Lister R, et al. (2011) Hotspots of aberrant epigenomic reprogramming in human induced pluripotent stem cells. *Nature* 471, 68-73.
15. Ruiz S, et al. (2012) Identification of a specific reprogramming-associated epigenetic signature in human induced pluripotent stem cells. *Proc Natl Acad Sci USA* 109, 16196-16201.
16. Kim K, et al. (2011) Donor cell type can influence the epigenome and differentiation potential of human induced pluripotent stem cells. *Nature Biotechnol* 29, 1117-1119.
17. Polo JM, et al. (2010) Cell type of origin influences the molecular and functional properties of mouse induced pluripotent stem cells. *Nature Biotechnol* 28, 848-855.
18. Kytala, A, et al (2016). Genetic variability overrides the impact of parental cell type and determines the iPSC differentiation potential. *Stem Cell Rep*, 6: 200-212.
19. Pastor, W.A., et al., TFAP2C regulates transcription in human naive pluripotency by opening enhancers. *Nat Cell Biol*, 2018. **20**(5): p. 553-564.

REVIEWERS' COMMENTS:

Reviewer #1 (Remarks to the Author):

"Vascular Progenitors Generated from Tankyrase Inhibitor-Regulated Naïve Diabetic Human iPSC Potentiate Efficient Revascularization of Ischemic Retina"
NCOMMS-19-18543

The authors have diligently worked to revise the manuscript, significantly improving its scientific rigor and clarity. Most of the original concerns have been thoroughly addressed, and the authors should be commended for this rigorous effort. However, a few concerns/suggestions remain, as specified below.

Figure 2. As requested, a new supplementary datasheet (Table S4) was created with densitometry quantitation tables of all Western blots shown in this manuscript (normalized to their actin controls using ImageJ software).

Comment - It would be best to include this information in graphical form rather than tabular form, even if in a supplementary figure. The table was not included in the returned manuscript file, but please make sure that at minimum, statistics are shown on the table.

Figure 3b-d: Please show statistics. 3b-c: Please include legends for graphs.

Comment - Figure 3B is only two repetitions, so statistical analyses could not be performed. An additional replicate should be performed such that this complies to Nature journal standards.

Fig. 4a, c, d, e: Please show statistics where appropriate.

Figure 4 has been completely revised. Statistical information has now been included in detail in the figure legend, and data has been re-graphed where relevant to show independent measurements of each assay ("n").

Comment - Figure 4B is only two repetitions, so statistical analyses could not be performed. An additional replicate should be performed such that this complies to Nature journal standards.

Figure 6d, e: Please show statistics, and label the top of the graphs with the time points. 6e: This panel is not cited in the main text

Figure 6 has been completely revised. Labeling of all figures and panels was done as recommended. All statistics are now shown in both the figures and figure legends.

Comment - Figure 6 is not present in the revised manuscript file, so it is difficult to assess these points. However, in the legend, multiple unpaired t tests were used. Would an ANOVA have been more appropriate if there are more than two groups? Again, this difficult to assess without the figure.

Please better introduce the ischemic retinopathy model in the main text. Most readers will not be familiar with the model, as this is a general journal.

Thank you for this great suggestion. (i.e., Fig. 6a).

Comment - Figure 6 is not included, so the schematic cannot be evaluated. However, judging from the manuscript's revised text, this point was addressed adequately.

Reviewer #2 (Remarks to the Author):

The author has basically worked on all of my questions and concerns except for immunostaining images. I am suspicious of KLF2 expression in naïve human PSC including LIF-3i iPSC. Authors claimed KLF2 is a nuclear pluripotency factor, but I can find very strong positive staining in the cytoplasm. So I asked authors to show western blotting with primed controls. Anyway, the expression of KLF2 and other naïve markers is not the main claim of this report. I recommend to delete the KLF2 image or to show that protein level is up-regulated in LIF-3i iPSC compared to primed PSC. I also point out that the staining pattern of TFCEP2L1 is unusual. If this is resolved, the paper will be accepted.

Reviewer #3 (Remarks to the Author):

The authors have done an excellent job of responding to my previous concerns. I have no further issues. The abstract, introduction, results and discussion have been edited to make the manuscript clearer and easier to understand. Pertinent prior studies have been included to help the reader appreciate the significance of this manuscript. I would like to congratulate the authors on a beautiful piece of work.

December 8th, 2019

RESPONSE TO REVIEWERS

REVISED RE-SUBMISSION (NCOMMS-19-18543B)

“Vascular Progenitors Generated from Tankyrase Inhibitor-Regulated Naïve Diabetic Human iPSC Potentiate Efficient Revascularization of Ischemic Retina”

We are grateful for the additional critiques of our Reviewers for improving the manuscript. Please note that this final version of the manuscript has the following major edits:

- 1) As per the Editor’s request, the abstract and text have been shortened to comply with journal standards of <150 words for the abstract and <5000 words for the main sections.
- 2) The figure for the RNA-Seq experiments (which were originally requested by Reviewer 2 as new experiments in the first revision of the paper) was moved from the Supplementary data section (previously Fig. S8), and is now a main figure (i.e., now **Figure 8**). We felt that the results of these bioinformatics experiments were too important to be “buried” in the Supplementary section. Because of this addition of a figure into the main section, the paper now has 10 total display items (figures), which is still within the maximum allowed number of display items for a *Nature Communications* Article.

Otherwise, below is a detailed point-by-point address of the remaining comments and critiques provided to us by our Reviewers.

Reviewer 1 Comments:

The authors have diligently worked to revise the manuscript, significantly improving its scientific rigor and clarity. Most of the original concerns have been thoroughly addressed, and the authors should be commended for this rigorous effort. However, a few concerns/suggestions remain, as specified below.

Figure 2. As requested, a new supplementary datasheet (Table S4) was created with densitometry quantitation tables of all Western blots shown in this manuscript (normalized to their actin controls using ImageJ software). Comment -It would be best to include this information in graphical form rather than tabular form, even if in a supplementary figure. The table was not included in the returned manuscript file, but please make sure that at minimum, statistics are shown on the table.

Hopefully, our Reviewer now has access to the new supplementary datasheet (**Table S4**) that is submitted for final publication. Please note that in this supplementary file (which is a multiple-page EXCEL spreadsheet) there are not only densitometry quantitation of all Western blots **in tabular format** (normalized to their actin controls using ImageJ software) but each spreadsheet page also includes embedded bar graphs for each of the densitometry tables, as well as the associated

embedded WB blot band images. Where relevant, *p* value statistics were also included and shown in the spreadsheet page (e.g., for Fig. 2b,2e). Please also note that as required for publication in *Nature* journals, we have included the images of all raw, uncropped Western blots presented in this paper in a zip-compressed 'Source Data' supplementary file. Thank you.

Comment - Figure 3B is only two repetitions, so statistical analyses could not be performed. An additional replicate should be performed such that this complies to Nature journal standards.

The experiments in the graph of differentiation kinetics of DhiPSC (previously shown in Figure 3b) were not designed to be powered to discern differences in TRA, CD34 expression, etc, but were intended to be a concept figure that accompanied the vascular differentiation schematic (*i.e.*, previous Figure 3a). The point of that differentiation protocol and kinetics schematic was:

1) to show that primed and naïve DhiPSC have similar patterns and timelines of differentiation kinetics (*i.e.*, similar CD31⁺CD146⁺ VP kinetics of generation, as was also demonstrated for non-diabetic primed and naïve hiPSC in our *Development* and *Circulation* papers), and,

2) to demonstrate why we chose day 8-10 of vascular differentiation for subsequent quantitative CD31⁺CD146⁺ VP analyses (*i.e.* Figures 3e, 3e) in primed and naïve DhiPSC (*i.e.*, because preliminary kinetic experiments revealed that days 8-10 of vascular differentiation were the peak times of CD31⁺CD146⁺ cell generation in our differentiation protocol for both primed and naïve hiPSC).

Thus, knowledge of these differentiation kinetics of our vascular protocol were use to determine the subsequent quantitative CD31⁺CD146⁺ VP measurement (and other vascular lineages) at days 8-10 of differentiation.

Those experiments were shown in **Figure 3**, were powered with enough replicates from multiple biologically independent hiPSC lines to determine the statistical significance of increased vascular cell generation of naïve vs primed hiPSC. We have moved this background differentiation kinetics schematic to serve as a supplementary figure (**Fig. S3**). Instead, we now show only the quantitative vascular differentiation experiments at the optimal day 10 of differentiation for nondiabetic hiPSC (**Fig 3a**) and diabetic hiPSC (**Fig. 3c**) to place this important data front and center into a revised main **Figure 3**.

Thanks for this important critique which helps clarify the presentation of our data.

Figure 4 has been completely revised. Statistical information has now been included in detail in the figure legend, and data has been regraphed where relevant to show independent measurements of each assay ("n"). Comment - Figure 4B is only two repetitions, so statistical analyses could not be performed. An additional replicate should be performed such that this complies to Nature journal standards.

Thank you. An additional replicate from an additional biologically independent hiPSC line is now included in this figure, and is presented as a supplementary Fig. S5c in this revised manuscript. Although the data and SEM show increased mean proliferation of naïve VP in this in vitro assay, unfortunately, the *p* values were not <0.05, are not labeled, and no significance is claimed.

Comment - Figure 6 is not present in the revised manuscript file, so it is difficult to assess these points. However, in the legend, multiple unpaired t tests were used. Would an ANOVA have been more appropriate if there are more than two groups? Again, this difficult to assess without the figure. Please better introduce the ischemic retinopathy model in the main text. Most readers will not be familiar with the model, as this is a general journal. Thank you for this great suggestion. (i.e.,

Fig. 6a). Comment - Figure 6 is not included, so the schematic cannot be evaluated. However, judging from the manuscript's revised text, this point was addressed adequately.

Hopefully, our Reviewer now has access to the **Fig. 6** with the appropriate statistical tests demonstrated. Please also note that in response to our Reviewer's original recommendations, the revised **Fig. 6** included a new schematic that illustrates the I/R model (*i.e.*, Fig. 6a). We also expanded the text on pp. 9-10 where we introduce the model, and provided more background, explanation, and references of the rationale for use of this I/R model that we employed to test the functionality of naïve vs primed DVP.

Many thanks to our Reviewer 1 for all the helpful and detailed critiques which improved our manuscript.

Reviewer 2 Comments:

The author has basically worked on all of my questions and concerns except for immunostaining images. I am suspicious of KLF2 expression in naïve human PSC including LIF-3i iPSC. Authors claimed KLF2 is a nuclear pluripotency factor, but I can find very strong positive staining in the cytoplasm. So I asked authors to show western blotting with primed controls. Anyway, the expression of KLF2 and other naïve markers is not the main claim of this report. I recommend to delete the KLF2 image or to show that protein level is up-regulated in LIF-3i iPSC compared to primed PSC. I also point out that the staining pattern of TFCEP2L1 is unusual. If this is resolved, the paper will be accepted.

As recommended by Reviewer 2, and because these immuno-stains are not at all central to the main findings of this paper, we revised **Fig. 2** and deleted the immuno-stain images for KLF2 and TFCEP2L1. Also, please note for future reference, that in our hands, and with the antibody we used, the staining of TFCEP2L1 in N-hPSC appears primarily nucleolar, as previously described: <https://www.proteinatlas.org/ENSG00000115112-TFCEP2L1/cell> .

In any case, we now include an additional supplementary **Table S6**, which summarizes the most significantly ($p < 0.05$) differentially-expressed genes in both normal N-hPSC and diabetic N-DhiPSC vs their primed hPSC counterparts ($n=8$ hPSC lines). This data is compiled from the new bulk RNA-Seq experiments that were requested from this Reviewer in our first revision, and are now presented as a main **Figure 8**.

Please note from the data in **Table S6** that both normal and diabetic N-hPSC significantly over-express *KLF2* transcripts relative to primed hPSC (*i.e.*, N-hPSC: log₂ fold difference *KLF2* = 3.29; *i.e.* 9.8-fold higher levels than primed normal isogenic hPSC ($p < 10^{-6}$), and N-DhiPSC: log₂ fold difference *KLF2* = 1.5, *i.e.* at 2.8-fold higher levels than isogenic conventional diabetic D-hiPSC). Please also note from this table that both normal N-hPSC and diabetic N-DhiPSC both over-express a large panoply of naïve-specific transcripts at very high significances ($p < 0.001$) at levels ~5-100-fold higher than primed isogenic counterparts, including *DNMT3L*, *SP5*, *PRDM14*, *DPPA3* (*Stella*), *TBX3*, *NR5A2*, *KLF5*, *KLF4*, *KLF11*, *NANOG*, and many other naïve pluripotency-specific genes.

The full RNA-Seq data has been uploaded into GEO for accession after publication, as per Nature guidelines.

Thank you, once again, for the rigorous review of this manuscript by Reviewer 2.

Reviewer 3 Comments:

The authors have done an excellent job of responding to my previous concerns. I have no further issues. The abstract, introduction, results and discussion have been edited to make the manuscript clearer and easier to understand. Pertinent prior studies have been included to help the reader appreciate the significance of this manuscript. I would like to congratulate the authors on a beautiful piece of work.

We are sincerely appreciative of Reviewer 3 for a thorough review, and for providing us with detailed and constructive critiques which helped make this a significantly better manuscript.